# Increasing control over biomineralization in conodont evolution

Bryan Shirley[1,2], Isabella Leonhard[3,4], Duncan J. E. Murdock[5], John Repetski[6], Przemysław Świś [4,7], Michel Bestmann[8], Pat Trimby[9,10], Markus Ohl[2], Oliver Plümper [2], Helen E. King [2] & Emilia Jarochowska [2] ✉

Vertebrates use the phosphate mineral apatite in their skeletons, which allowed them to develop tissues such as enamel, characterized by an outstanding combination of hardness and elasticity. It has been hypothesized that the evolution of the earliest vertebrate skeletal tissues, found in the teeth of the extinct group of conodonts, was driven by adaptation to dental function. We test this hypothesis quantitatively and demonstrate that the crystallographic order increased throughout the early evolution of conodont teeth in parallel with morphological adaptation to food processing. With the *c*-axes of apatite crystals oriented perpendicular to the functional feeding surfaces, the strongest resistance to uniaxial compressional stress is conferred along the long axes of denticles. Our results support increasing control over biomineralization in the first skeletonized vertebrates and allow us to test models of functional morphology and material properties across conodont dental diversity.

Vertebrates are one of the few lineages of skeletonized animals that form their skeletal tissues out of the phosphate mineral apatite[1] and the one that has achieved the highest disparity. Composite organic-apatite biomineral forming vertebrate skeletons range through internal bone scaffolds, body armour, teeth, and scales[2,3]. These versatile, remodelable tissues allowed a diversity of new feeding and protective structures, which triggered an evolutionary escalation of complex trophic interactions[4]. All other major, widespread animal groups build their hard tissues from softer minerals: calcite, aragonite, or robust organic polymers[1]. Vertebrates stand out in terms of the mineralogy of their skeletons and the disparity of structures used for feeding, locomotion, and defense. This disparity coincides with the outstanding evolutionary and ecological success of this group.

The earliest phosphatic skeletal tissues found in the evolutionary history of vertebrates are conodont teeth, which evolved in parallel to all other vertebrate teeth. Donoghue[5] put forward the hypothesis that

the biomineral structure and properties of conodont teeth evolved as an adaptation to dental function. This hypothesis is the basis of our approach here: hypotheses on the function of a skeletal tissue can be tested by predicting its material properties from its crystallographic structure. However, conodont skeletons are very heterogeneous, and in situ studies relating crystal orientations to individual functional surfaces have been very limited. We offer a quantitative test of the hypothesis by Donoghue[5] by quantifying the degree of an organism's control over the crystallographic order in conodonts.

Conodonts are paramount to understanding the origins of skeletal evolution thanks to their long evolutionary history and morphological diversity[6,7]. These marine vertebrates bore apparatuses composed of morphologically and functionally differentiated teeth (also referred to as elements). These elements consisted of composite materials formed by organic matter and hydroxyapatite $(Ca_{10}(PO_4)_6(OH)_2)$[8–10]. Such composite materials also form vertebrate

[1]Fachgruppe Paläoumwelt, Friedrich-Alexander-Universität Erlangen-Nürnberg, Erlangen, Germany. [2]Department of Earth Sciences, Utrecht University, Utrecht, Netherlands. [3]Department of Palaeontology, University of Vienna, Vienna, Austria. [4]Institute of Evolutionary Biology, Biological and Chemical Research Centre, Faculty of Biology, University of Warsaw, Warsaw, Poland. [5]Oxford University Museum of Natural History, Oxford, UK. [6]US Geological Survey-Emeritus, MS 926 A National Center, Reston, USA. [7]Department of Chemical and Geological Sciences, University of Modena and Reggio Emilia, Modena, Italy. [8]Department of Geology, University of Vienna, Vienna, Austria. [9]Oxford Instruments, High Wycombe, UK. [10]Present address: Carl Zeiss Ltd., Cambridge, UK. ✉e-mail: e.b.jarochowska@uu.nl

bones, teeth, and mineralized scales. Conodont elements typically range from 0.1 mm to 2 mm in length, with rare giants reaching over 13 mm[11]. The morphology of their elements ranges from simple cone-like forms to molar-like structures with complex topography allowing accurate occlusal (mastication)[12,13].

The oldest conodonts are represented by the paraphyletic grade of paraconodonts, which appeared in the Cambrian Period and were characterized by poorly mineralized phosphatic tissues rich in organic matter. Paraconodonts were succeeded by the more derived group, euconodonts, which dominated the lineage from the late Cambrian (ca. 490 Ma) until the extinction of conodonts at the end of the Triassic Period (ca. 200 Ma). Euconodonts were distinguished by hypermineralized crown tissues, which formed on top of less mineralized basal bodies with homologous structures to paraconodont elements. Crown tissues formed the functional part of the dental organs, which participated in the mechanical digestion of prey. Euconodont crown consists of two types of tissues: the white matter, unique to conodonts, and the enamel-like hyaline tissue[7,14]. The hyaline tissue is widely used by geochemists for paleoclimatic reconstructions thanks to its outstanding resistance to diagenesis[15]. This tissue grew by apposition throughout the life of the organism. Its complex developmental mechanism permitted conodonts to achieve enormous morphological diversity[16,17]. The known range of morphologies reflects adaptation to a broad range of trophic niches. For some conodont lineages, a trend towards developing complex surface topographies for stress dissipation and efficient food use has been reconstructed[18,19]. Other taxa evolved unparalleled sharpness interpreted as a mechanism of efficient food shearing and slicing[20]. Trophic diversification of conodonts is supported by distinct, species-specific geochemical values of proxies for trophic partitioning[21].

In this study, we reconstruct the selective pressure that acted upon the earliest vertebrate biominerals and shaped early conodont evolution by examining the link between their ultrastructure and functional morphology.

Here we use Electron Backscatter Diffraction (EBSD) to characterize the crystallographic preferred orientation, in the following referred to as crystallographic texture, of conodont hyaline tissue across a range of morphologies and adaptations to food processing. Based on the textural development through conodont evolution, we infer structural adaptations using criteria linking crystallographic texture with resistance to compressional stress. The *c*-axis of an apatite crystal is the most resistant to uniaxial compressional stress[22] and, thus, the most resilient to breakage when uniaxial compression is applied (Fig. 1). Consequently, a stronger *c*-axis alignment would disperse stress throughout the tooth better and allow for a stronger bite and reduced risk of breakage[23,24].

Crystallographic texture can be affected by diagenesis, chemical processes taking place after the death of the organism, potentially overprinting the patterns formed in vivo[25,26]. In conodonts, the degree of diagenetic alteration is commonly estimated using Conodont Alteration Index (CAI), a proxy for the temperature to which elements had been subjected, and – thus – for burial conditions[27]. For this study, we selected only conodonts with the lowest CAI values of 1.0 to 1.5[28]. Diagenesis in conodonts has been shown to largely emphasize existing crystal orientations[29], which would have limited impact on our analysis. To account for any potential differences between specimens that might be due to diagenetic histories, we acquired Raman spectra from the same areas in which crystallographic texture was analyzed. The position and the width of the $v_1$-$PO_4^{3-}$ Raman band is known to vary in response to incorporation of non-phosphate ions in the phosphate structural site[30], with lower values of the position (larger redshift) correlating with diagenetic alteration[15].

EBSD results are quantified using the pole density function (PDF), which is calculated based on the relative frequencies of lattice plane orientations. The pole density function is visualized as pole figures: two-dimensional stereographic projections representing the crystal orientations projected onto a 3D sphere. While a great visual representation of texture, pole figures do not offer a quantitative measure of crystal alignment[31]. Such measure can be obtained using the Orientation Distribution Function (ODF, Eq. 1)[32,33]. ODF provides a quantitative measure of crystal orientations not only within a single sample, but can also be compared between samples. From ODF, several texture indices can be calculated, expressing in a scalar value the degree of co-alignment of crystals. Here we apply these indices to our datasets and, based on their comparison, focus on the Texture Index (TI, Eq. 2)[33,34] as allowing the best resolution for the textures analyzed here.

With the application of the ODF method and calculation of TI, crystal textures can be compared across tissues, providing a proxy for the level of biomineralization control. We characterize the crystal texture of six conodont taxa representing increasing morphological complexity (Supplementary Fig. 1), reflecting a range of adaptations to dental function. Two morphologically simple coniform taxa are *Proconodontus muelleri* (Supplementary Fig. 1A), one of the earliest euconodonts with comparatively limited crown tissues, and *Panderodus equicostatus* (Supplementary Fig. 1B), a morphologically simple representative of euconodonts with fully mineralized and differentiated skeletal tissues[7]. Conodonts with complex morphologies are: *Bispathodus* cf. *aculeatus* (Supplementary Fig. 1C) and *Wurmiella excavata* (Supplementary Fig. 1D), both representing lineages for which biomechanical studies infer adaptations to food slicing[20], and platform-bearing *Tripodellus gracilis* (Supplementary Fig. 1E) and *Palmatolepis* sp. (Supplementary Fig. 1F). We test for increasing

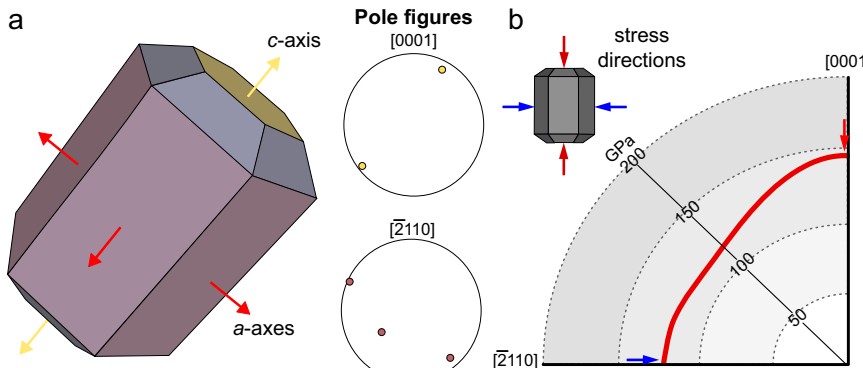

**Fig. 1 | 2D representation of 3D crystal orientations using pole figures. a** A hexagonal crystal, such as apatite, with the *c*-axis <0001> and *a*-axes <10–10> marked and their respective appearance in pole figures. **b** A visual representation of the change in Young's modulus of fluoroapatite in the (10–10) plane, i.e. relative to the crystal direction. The red and blue arrows illustrate shear stress acting in different directions of crystal orientation. Modified from[22].

ultrastructural adaptation by comparing TI values along the axes of mastication movement (axes of occlusal[35]) of individual denticles, i.e., along the directions subject to the highest compressional stresses.

In this work, we show that the evolution of the hypermineralized tissues in the teeth of conodonts, representing the earliest skeletonized vertebrates, was an adaptation at the ultrastructural level. Apatite, which forms the mineral phase of vertebrate skeleton, has the highest elasticity along its mineralogical *c*-axis. By measuring in situ distributions of crystals across six conodont taxa, we identify that in more derived conodonts the *c*-axes of apatite in their teeth are more aligned in the inferred direction of dominant stress during biting. Since conodonts and other early vertebrates are extinct, the mechanism through which they acquired trophic differentiation, have been so far reconstructed from their functional morphology, distribution, and geochemical proxies. Here, we show that crystallographic preferred orientations in a biomineral allow reconstruction of its material properties and infer its function in the living organism.

## Results

### Crystallographic texture

The orientations of *c*-axes are mostly parallel to the long axes of the denticles and cusps. In the earliest diverging taxon, *Pro. muelleri*, the *a*-axes <11−20> show a girdle distribution resulting from gradual rotation around the *c*-axis, visible as a color gradient in Fig. 2. A similar distribution but with a lower dispersion around the *c*-axis was found in the element of the more derived coniform conodont, *Pan. equicostatus* (Fig. 2b). The range of rotation around the *c*-axis decreases systematically within each specimen (see further for the quantification), leading to a constrained distribution with the *a*-axes of crystals showing very little dispersion in *W. excavata* (Fig. 2c), T. *gracilis* (Fig. 2e) and the most derived taxon, *Palmatolepis* sp. (Fig. 2f). In *B*. cf. *aculeatus* (Fig. 2d), the crystals formed three distinct clusters (of the *a*-axis) in terms of their angles of rotation around the *c*-axis.

In biominerals, grains are areas with similar crystal orientations and appear in the same color on a crystallographic orientation map, even though they may consist of multiple nanocrystals[36]. Grain sizes can be used to characterize a material, and they impact its properties. Here, however, grain sizes have not been calculated, as this operation requires grain reconstruction following algorithms that rely on constant misorientation angles between adjacent grains. We demonstrate (Supplementary Fig. 2) that the misorientations vary between areas within one tissue depending on their position with respect to functional directions and the compressional and shear stresses acting in these directions. Thus, standard approaches to grain analysis, which rely on constant misorientation angle thresholds, may not be suitable for complex biominerals, where these thresholds change during evolution and even within one tissue type. Furthermore, in abiotic crystalline materials, the long axes of grains commonly correspond to the crystallographic *c*-axis of the mineral. This is not always the case in biological tissues[37]. Although grains have not been computed using an algorithm, elongated clusters of points with similar crystallographic orientations can be distinguished here, especially in *Pro. muelleri*. In this specimen, in areas where such clusters are visible, their long axes do not correspond to crystallographic *c*-axes <0001 >.

### Spread of orientations

A decreasing tendency in the spread of orientations is visible in pole figures from the least derived taxon *Pro. muelleri* (Fig. 2a) to the most derived ones, *T. gracilis* and *Palmatolepis* sp. *Pro. muelleri* stands out with the lowest TI values ranging from 4 to 48 (mean 12, Fig. 3a). All five taxa with fully developed crown tissues overlap (Fig. 3), but the coniform *Pan. equicostatus* occupies a much smaller range of TI values (17 to 92, mean 29), compared to "complex" conodonts, *B*. cf. *aculeatus* (24 to 71, mean 36) and *W. excavata* (9 to 67, mean 35). Taxa with elements forming wide platforms, considered highly adapted to

mastication[18], have the highest TI values of all: *T. gracilis* (38 to 75, mean 57) and *Palmatolepis* sp. (13 to 92, mean 68).

In heterogeneous materials such as biogenic apatite, measures of spread depend on the size of the analyzed area (Supplementary Fig. 3). To eliminate sampling bias, random subsets were drawn from the measured areas and, for each subset, ODF and TI were calculated (Fig. 4). The smaller range of values for *Pan. equicostatus* cannot be explained by the smaller area of the element exclusively, as subsampling from other euconodont taxa yielded higher TI values for comparable areas (Fig. 4).

Because TI is quite sensitive to the area (or number of EBSD data points), we evaluated the robustness of our conclusions by repeating the analysis using another measure of the strength of preferred orientation proposed in the literature, the M-index[38,39]. It is calculated from the misorientation angles between uncorrelated grain pairs. Additionally, we evaluated the sharpness of the texture in the directions of *c*- and *a*-axes using the pole figure texture index *pfJ*[31] (Fig. 4, Supplementary Fig. 4, and Supplementary Table 1). TI followed by *pfJ* showed the best sensitivity to differences in texture between our samples and allowed resolving the differences best.

### Evaluation of potential diagenetic influence

Peak positions of the $v_1$-$PO_4^{3-}$ band obtained through Raman analyzes overlap between studied conodonts (Fig. 5a). They range from 963.67 to 964.26 cm$^{-1}$ for the band position and from 4.28 to 5.67 cm$^{-1}$ for the full width at half maximum (FWHM; Fig. 5b)[28]. *Palmatolepis* sp. is the only exception, with lower values for both band position (mean 963.88 cm$^{-1}$) and FWHM (mean 4.49 cm$^{-1}$, both n = 5).

Diagenetic alteration such as uptake of elements post-mortem has been reported to proceed on the outer surface of the elements[40]. Therefore, we compared also the CAI values and distance from the center in each specimen against the position of the $v_1$-$PO_4^{3-}$ band. No relationship between these variables and $v_1$-$PO_4^{3-}$ band position was found (Supplementary Fig. 5).

Zhang et al.[15], noted that higher values for the band position (lower redshifts) correspond to samples with little diagenetic alteration. Thus, if the specimens studied here had experienced different degrees of diagenesis, a high spread of the band position would be expected. Figure 5 shows the relationships between the peak center at maximum intensity (PCMI) and FWHM of the $v_1$-$PO_4^{3-}$ band in our specimens compared with those reported by McMillan and Golding[41], Rantitsch et al.[42] and Zhang et al.[15], which were the only published datasets available at this point.

In all datasets, the FWHM of the peak decreased with an increasing position of the band center (Fig. 5). The datasets by both McMillan and Golding[41] and Rantitsch et al.[42] showed high variability in the position and widths of the $v_1$-$PO_4^{3-}$ band. Model II regression was fitted to the relationship for these two datasets, producing slopes equal to −0.92 ($R^2 = 0.63$, N = 41) and −1.30 ($R^2 = 0.90$, N = 153), respectively (Fig. 6). Both slope and intercept differed significantly between these two datasets at a confidence level of 0.05. In contrast, our dataset and that by Zhang et al.[15] showed strong clustering close (Zhang et al.) or directly on the regression line fitted to the values measured by Rantitsch et al.[42]. Compared to the other studies, positions of the $v_1$-$PO_4^{3-}$ band measured for our specimens showed consistently high values with a mean of 964.00 (median 963.99, standard deviation 0.13, n = 30).

## Discussion

### Diagenesis

Diagenesis refers to chemical and physical alterations altering the original composition and mineralogy of, in this case, fossils. It can take place in the water column even during the life of the organism, in the immediate post-mortem interval when the fossil becomes part of the sediment components on the seafloor, or during burial in the

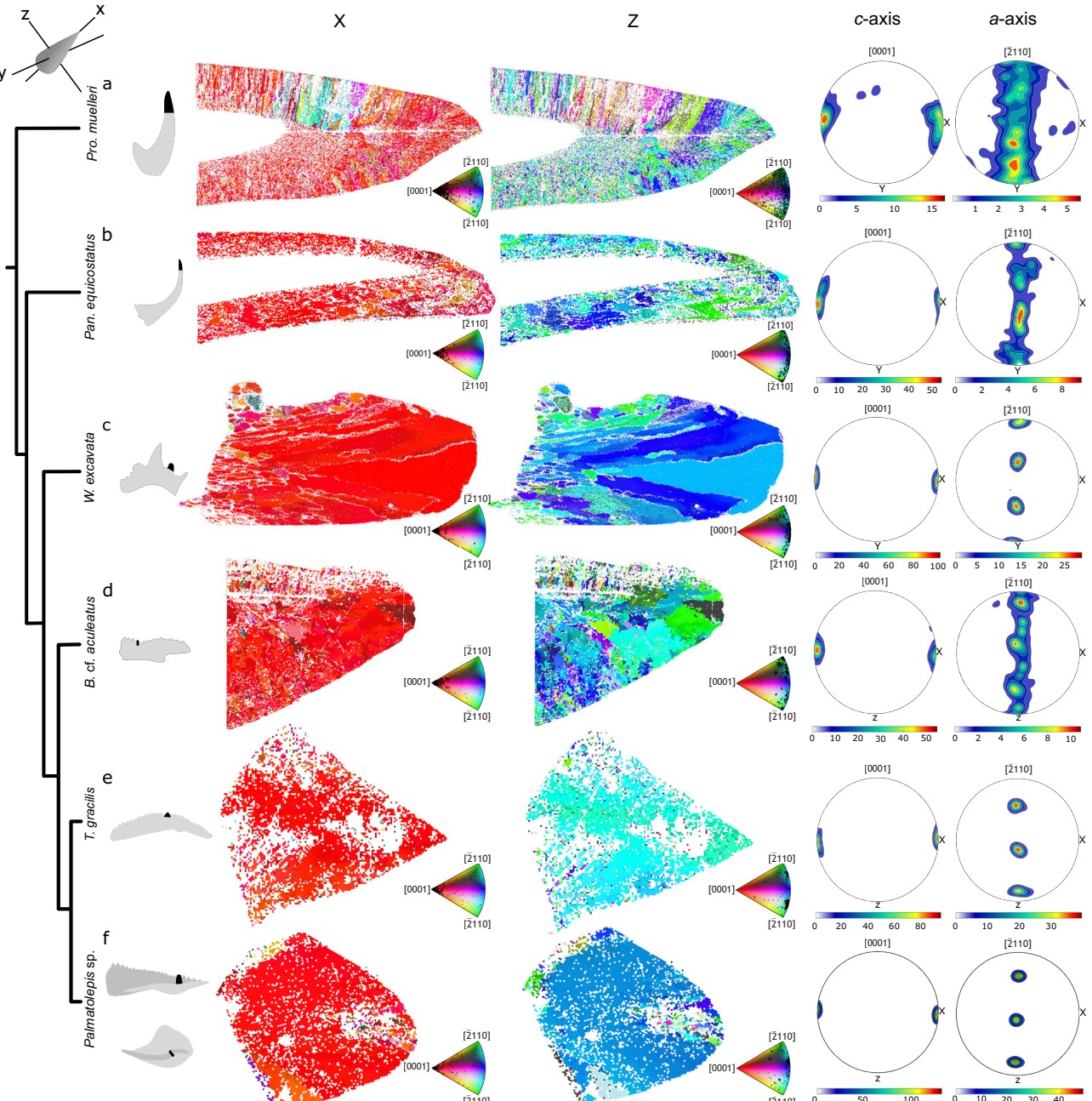

**Fig. 2 | Crystallographic textures of conodonts.** From left to right: A cladogram of relationships (adapted from[7,55]) of taxa with silhouettes showing the location (black) in the element. Colors in the crystallographic orientation maps are defined in relation to reference directions determined by biological orientation in a tooth and illustrated in the top left corner. The X axis is defined by the long axis of the denticle and approximates the occlusal axis. In the maps on the left, the colors show crystallographic orientations relative to the X direction, and in the maps on the right, relative to the Z direction. In the lower right corner of each map, the key to the colors used on the map is provided by inverse pole figures (IPFs), in which the orientation of the specimen coordinate system (X, Y, Z) is projected into the crystal coordinate system[87]. Pole figures (right) show orientation distributions corresponding to the maps on the left. The values shown are Multiples of Uniform Density (MUD), which describes the preferred orientation. Source Data for this figure is available at https://osf.io/m26qa/.

sediment, which – on longer time scales – can be associated with elevated temperatures and pressure. For calcareous biominerals, experimental studies have established the effects of typical diagenetic pathways observed in marine environments[26,43,44]. These effects typically include neomorphism, i.e. recrystallization, commonly into larger grains, homogenization of crystal orientations, and chemical exchange with fluids present during diagenesis. These effects depend on the stability of the original mineral, the porosity – and thus exchange surface area – and the contents of organic matter which can act as conduits for the exchange of water and ions. Apatite is more stable (i.e.

less soluble) in seawater than calcite or aragonite, minerals most commonly used by organisms to build skeletons[1]. Yet highly porous bioapatite tissues, such as bone and dentine, are still known to be susceptible to substantial diagenetic changes[45,46]. Conodont crown and its analog tissue, enamel, are the least susceptible tissues to diagenesis[9] owing to their low porosity[47] and low organic contents, but even in these tissues diagenetic exchange of ions and isotopes[9,40] and changes in crystallinity[42] can occur. As there are no other studies yet that would assess the crystallographic texture of fossil bioapatite quantitatively, no insights are available yet as to how diagenesis might

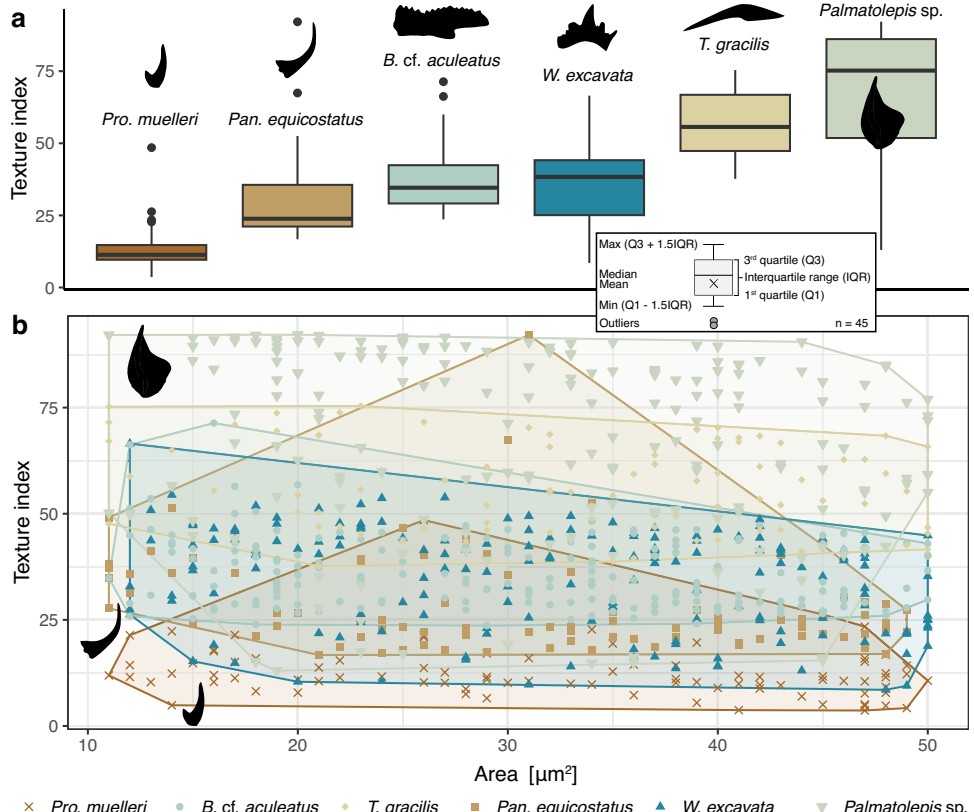

**Fig. 3 | Texture Index (TI)[31] values in conodont crown tissues. a** Distributions of TI values per taxon. The means between taxa are significantly different (Kruskal-Wallis test, $\chi^2 = 494.13$, 5 degrees of freedom, p-value $< 2.2 \times 10^{-16}$). Corrected pairwise comparisons and confidence intervals are reported in Supplementary Tables 3 and 4, respectively. **b** Relationship between TI and the area with convex hulls, based on the same areas as shown in Fig. 2. Each plot is based on N = 45 measurements per specimen. Source Data for the figure can be found at https://osf.io/yfk3x.

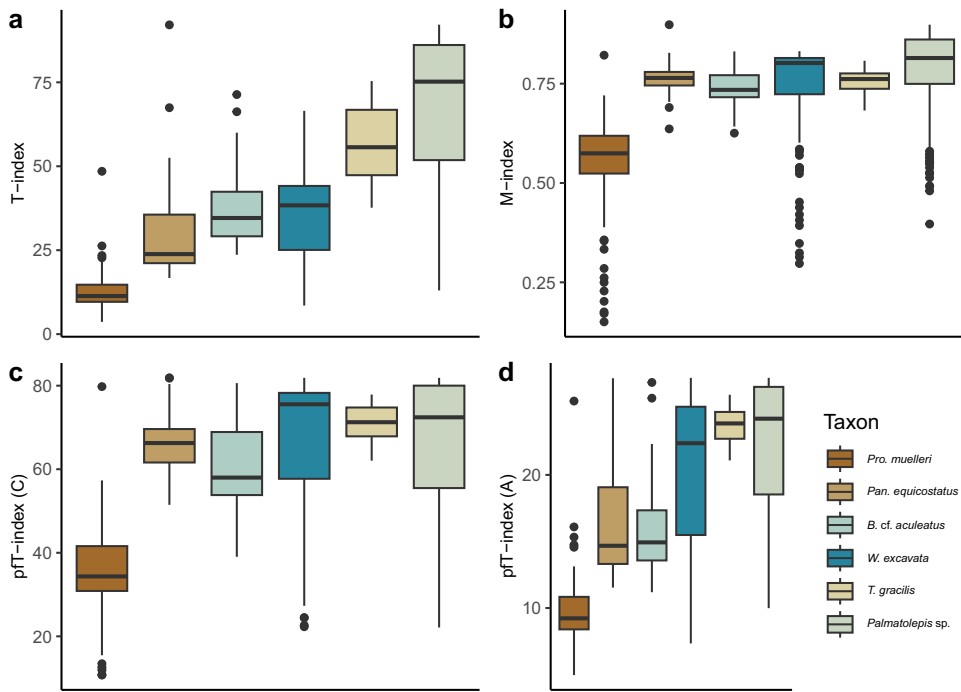

**Fig. 4 | Measures of orientation spread for each EBSD map in Fig. 2. a** Texture Index (TI), **b** M-Index and **c**, **d** pole figure texture index *pfJ* for **c**) the *a*-axis and **d**) *c*-axis. For boxplot legend, see Fig. 3. Source Data can be found at https://osf.io/yfk3x.

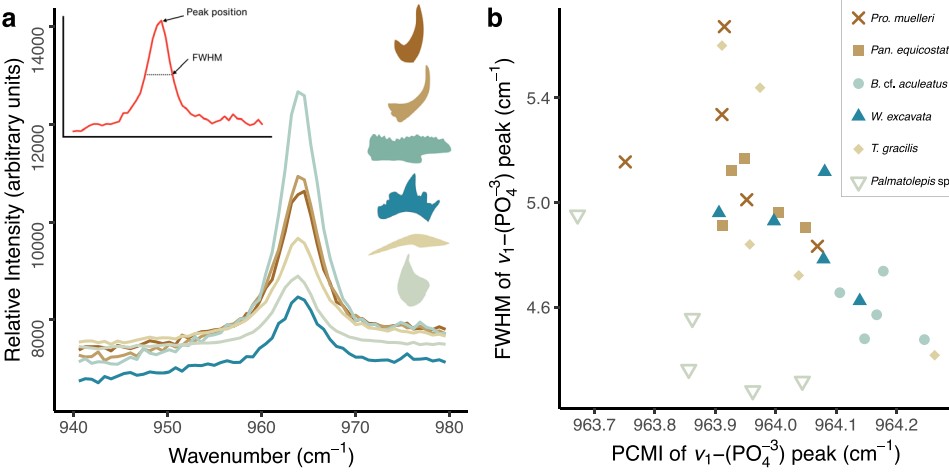

**Fig. 5 | Peak positions and full width at half maximum (FWHM) of the $v_1$-$PO_4^{3-}$ band obtained in Raman analyzes. a** Example spectrum for each respective taxon in this study, representative of the five measurements made on each specimen. **b** Variability of the peak parameters across the five measurements for each taxon. Source Data for Fig. 5a can be found at https://osf.io/smhjr and for Fig. 5b at https://osf.io/cgexs.

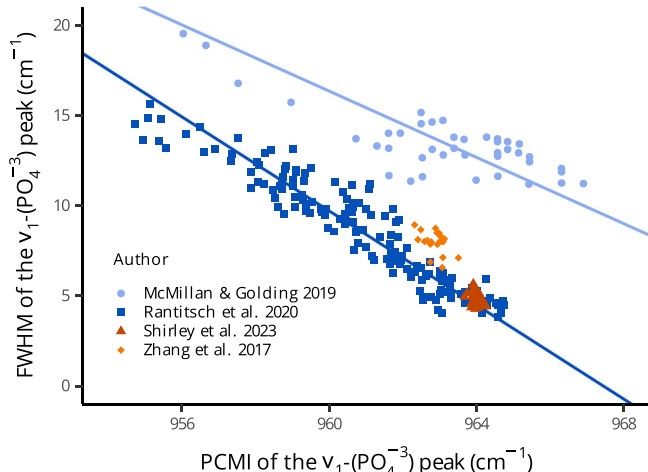

**Fig. 6 | Peak center at maximum intensity (PCMI) and full width at half maximum (FWHM) of the $v_1$-$PO_4^{3-}$ band in the Raman spectrum of apatite in this study (dark red triangles) compared with literature data from conodonts.** Fitted lines represent model II linear regression. Source Data for Fig. 6 can be found at https://osf.io/h85k4.

affect our conclusions. In the absence of such benchmark, we compared the peak positions of the $v_1$-$PO_4^{3-}$ band in the apatite across the specimens and with other fossil bioapatites using Raman spectroscopy. Compared to other datasets spanning a wide range of taxa and diagenetic environments, all our specimens showed remarkably similar parameters, supporting our original choice of specimens with the lowest expected diagenetic alteration. Based on this, we propose that differences observed in crystallographic textures between conodonts discussed here could not arise through diagenesis, as the specimens underwent similar diagenetic pathways. However, a systematic study of the diagenetic alteration of crystals in conodont elements is needed.

### The crystallographic texture of conodont denticles is an adaptation to dental function

We quantified crystallographic orientations, interpreted them using known relationships between crystallographic texture and material properties[48,49], and found support for Donoghue's hypothesis[5] that the biomineral structure of conodont teeth evolved as an adaptation to dental function. This hypothesis, however, was proposed based on observations of grains, also referred to as crystallites, which are visible as differential topography in broken or etched sections. In these sections, grains are elongated prisms oriented radially from the growth (and occlusal) axis. Such orientation suggested the strongest resistance to compressional stress acting upon the element along the occlusal axis. Here, however, we show that $c$-axes are perpendicular to grain elongation. In (hydroxy)apatite, elasticity differs substantially between the crystal directions and the highest Young's modulus is found in the $c$-axis direction (Fig. 1)[22]. Thus, the strongest resistance to deformation under uniaxial compressional stress is conferred along the long axes (X-axis) in single-cone elements or of the denticles in denticulate conodont elements.

Testing of the hypothesis on functional adaptation of conodont ultrastructure has not been possible until recent improvement in in situ quantification of crystallography of bioapatite, which is sensitive to the electron beam. Until now, the only successful EBSD study on conodonts[47] obtained EBSD diffraction signal from the white matter in two taxa, focusing on the crystallinity of this tissue without analyzing the implications for material properties of the elements. Thanks to the quantitative orientation data obtained from the hyaline tissue of conodont dental elements, we identify two quantifiable criteria for testing this hypothesis: (1) parallel orientation of $c$-axes to the occlusal surface, and (2) the degree of crystallographic order expressed using the Texture Index (TI). Our test is limited to six taxa spanning a wide range of morphologies, geological ages, and evolutionary lineages, but – within this limited set – the first criterion is fulfilled in all studied taxa, and the second, the degree of crystallographic order, increases in more derived taxa characterized by a higher degree of morphological adaptation to dental function. Furthermore, we examined additionally that the degree of crystallographic order does not differ between multiple denticles from an individual specimen substantially (Fig. 7).

We observe that the range of rotation around the $c$-axes decreases along the evolutionary tree of the studied species, indicating strict control on the rotation and thus orientations of $a$-axes in the most derived taxa *T. gracilis* and *Palmatolepis* sp. Owing to improvements in the processing of weak signal obtained from beam-sensitive materials, similar rotations around the $c$-axes have only recently been quantified in hydroxyapatite of human enamel[50], revealing that it persists even in most distantly related vertebrates.

Our findings are consistent with the observation of a high frequency of breakage perpendicular to the long axes of denticles[17,51]. The

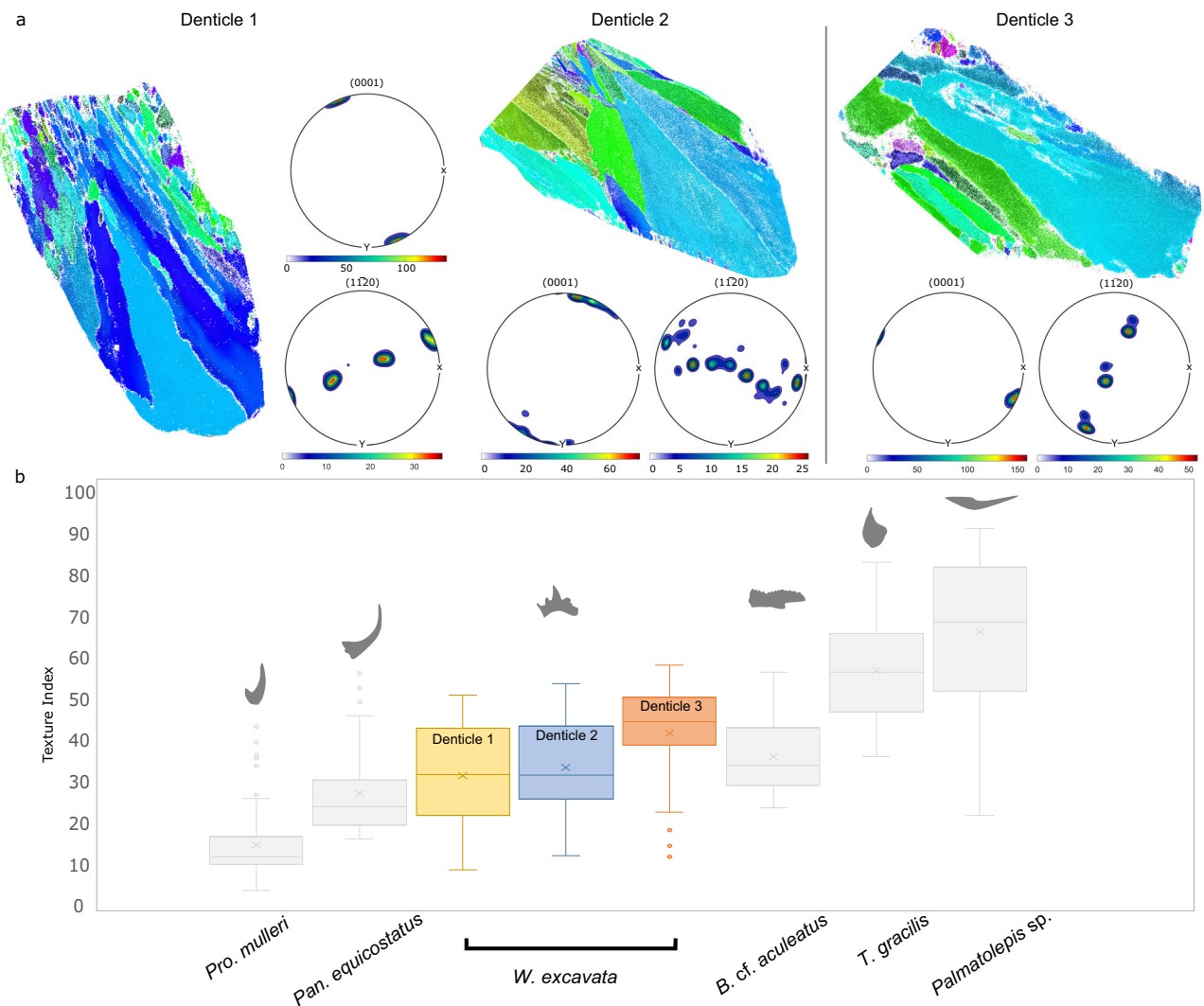

**Fig. 7 | Comparison of Texture Index values between denticles in *W. excavata*.**
**a** Orientation map colored relative to the X direction for denticles 1, 2 and 3. Pole figures show orientation distributions corresponding to the orientation maps.

**b** The Texture Index for denticle 1, 2 and 3 in comparison to the other taxa used in this study. Source Data for the figure can be found at https://osf.io/vnq6s/.

uniaxial stress acting upon a single cone element, a morphologically equivalent cusp or single denticle in a "complex" conodont element does not represent the entirety of stresses acting upon a "complex" element[20,52,53]. In "complex" conodonts, the occlusal may involve a multi-step cycle with stresses distributed differentially across the element[19,54]. This remains a major challenge in comparing the biomechanics of conodont elements across their entire evolutionary history[54]. However, for a single denticle, we assume that the dominant stress component can be simplified to uniaxial stress acting in the direction of its long axis. Expanding this analysis to the stress distribution acting upon the entire element would require considering the occlusion of each species individually, at the expense of being able to compare between taxa.

## Relationship between crystallography and ecology in studied taxa

The evolutionary relationships between conodont taxa have not been fully reconstructed[55,56], but available phylogenetic analyzes combined with functional morphology allows us to rank element morphologies from the least specialized feeding in *Pro. muelleri* to the most specialized ones in *Palmatolepis* sp. and *T. gracilis*. *Pro. muelleri* and *Pan. equicostatus* likely differed in their ecology from the predator and

scavenger trophic niches of "complex" conodonts[57]. The apparatus of *Pro. muelleri* contains elements with similar, comparatively simple morphologies[58]. Such a relatively undifferentiated and morphologically "primitive" apparatus is consistent with a non-specialized diet in contrast to those of a specialized predatory vertebrate. *Pan. equicostatus* is among the most advanced coniform taxa with differentiation of elements within the apparatus[52]. The body plan of this genus is consistent with the ability to swim freely above the sea bottom but not necessarily higher in the water column, i.e. nekto-benthic.[59] Frequent damage and repair of its elements indicate their use for mechanical food processing[16,60], but geochemical proxies place this taxon at the bottom of the trophic network compared to coexistent conodont taxa[21], further suggested a life habit restricted to the nekton-benthic realm. In the same trophic networks, *W. excavata* was placed at the top, consistent with its functional morphology. Finite element and sharpness analyzes indicate that *W. excavata* sliced tough, viscoelastic food[20], supporting its niche as a predator or scavenger. The more derived taxa *B.* cf. *aculeatus* and palmatolepidids are stratigraphically younger, thus, their trophic positions cannot be directly compared with the older taxa within the same ecosystems. Calcium isotopes indicated the position of *Palmatolepis* sp. as first-level consumers when compared with isotope ratios in macrophagous fish[61], but finite-

element analyzes of platform-bearing elements suggest that this morphology was an adaptation to increasing functional loads while processing food[19].

## Biological control

The presence of preferred crystallographic orientations is one of the criteria used to identify biominerals[36]. In extant models, the presence and direction of preferred crystallographic orientations reflects the stresses the biomineral is subjected to[62]. Thus, on evolutionary scales, the ability to produce biominerals with preferred crystallographic orientations is inferred to have adaptive value and is used to distinguish minerals with biological origin[63]. Our study proposes a quantification of the degree of crystal-level adaptation in vertebrate dental tissues at the evolutionary scale. We demonstrate that, across related taxa through the evolutionary history of a clade, the ultrastructure responsible for material properties shows an increased adaptation of the food processing mechanism. At the macroecological and macroevolutionary scale, this finding supports the hypothesis that the evolution of skeletal mouthparts is driven by competition for trophic niches[7].

Until now, biomechanical models of tooth enamel focused on humans and other extant mammals. These models rely on visible grain (also termed prism or crystallite) shapes. However, we demonstrate that grain shape orientations may not be representative of crystallographic orientations, which have been recently shown to be much more spread than the alignment of enamel grains[64]. Furthermore, a recent study on human enamel hydroxyapatite[50] showed systematic rotations around the c-axes that in grain reconstruction would be identified as grain boundaries, thus giving much smaller grains than present in the biomineral. Studies revealing controlled misorientations or gradual rotations indicate that precise biological control does not need to be manifested in a single preferred crystallographic orientation. A specific spread of orientations may convey mechanical properties[64] or arise from a specific biomineralization mechanism[65]. This is consistent with the observed range of textures here (Fig. 3), which reveals a structural diversity reflecting the broad range of conodont feeding mechanisms[21,54].

## Evolutionary significance

Apatite requires the limiting nutrient phosphorus, the acquisition of which is energetically costly. Vertebrates are the only widespread group using apatite for the secretion of skeletal tissues, and it has been postulated that apatite skeletons are associated with high-energy lifestyles of motile predators[1,66], requiring efficient food acquisition. Conodonts were the first among vertebrates to achieve high trophic and morphological disparity, and they contributed the greatest share of planktonekton at the onset of the Devonian Nekton Revolution[67]. The appearance of free-swimming animals with skeletonized, rapidly evolving food processing apparatuses has contributed to the evolutionary escalation of predation and defense mechanisms[4]. Increased biomineralization control shown here indicates that skeletal evolution was driven by selection for improved feeding function.

## Methods
### Specimens

All specimens are stored in the collections of Utrecht University (contact person E. Jarochowska), except for *Tripodellus gracilis*, which is stored in the collections of the Institute of Paleobiology, Polish Academy of Sciences. Elements prior to sectioning are shown in Supplementary Fig. 1. For anatomical notation see Purnell et al.[68] and for the phylogenetic position – Donoghue et al.[55].

***Proconodontus muelleri.*** A coniform aequaliform element (for anatomical notation see Müller, 1973) from the upper part of the Windfall Formation, at a section in Ninemile Canyon, Antelope Range, Eureka County, Nevada, USA (Supplementary Fig. 1A). It is from U.S. Geological Survey Cambrian-Ordovician sample locality USGS 12122-CO and is being stored under the accession number D000.000.389.576. The sample was collected on public land, and no permit is required for collection or export. The section was originally measured by J. Repetski, M.E. Taylor, and R.J. Ross, Jr. (all of USGS). The sample was collected by J.D. Loch (Central Missouri Univ.) and J.F. Taylor (Indiana University of Pennsylvania) and processed by J. Repetski. The conodont element was extracted from lime grainstone by processing in approx. 10% buffered acetic acid, sieved to 170 mesh (U.S. standard sieve size); the residue was not processed through heavy liquids. Age: late Cambrian, Stage 10; upper part of the *Eoconodontus* Biozone[69].

***Panderodus equicostatus.*** A coniform truncatiform element (for anatomical notation see Sansom et al., 1994) from Homerian (middle Silurian) Ternava Formation at Vrublivtsy, Ukraine (48°3703.90″ N; 26°4602.63″ E, Supplementary Fig. 1B). According to Ukrainian law, natural objects of value for the state's cultural and natural heritage, such as rare fossils, are prohibited from export, whereas common rock fragments and fossils are not included in this definition and can be collected on public land and exported without a permit. The sampled rock fragment was labeled V-19.25, for details of age and the locality, see ref. [70]. The rock sample was processed at GeoZentrum Nordbayern (Erlangen, Germany): it was dissolved in 7% buffered acetic acid according to the method of Jeppsson & Anehus (1995)[71], wet sieved, and the 63 μm – 1 mm fraction was separated in sodium polytungstate. Accession number D000.000.389.574.

**"Complex conodonts":** ***Bispathodus* cf. *aculeatus.*** A $P_1$ element of a polygnathid species from the scree in the natural outcrop of Famennian (Upper Devonian) limestone at Köstenhof, north-west of the hamlet of Elbersreuth in Frankenwald, southern Germany (Tragelehn & Hartenfels[72]; Supplementary Fig. 1C). It was collected on public land. No permits are required for collecting and exporting common fossils from Germany[73]. The rock sample was processed at GeoZentrum Nordbayern (Erlangen, Germany): it was dissolved in 7% buffered acetic acid according to the method of Jeppsson & Anehus (1995)[71], wet sieved, and conodonts were picked from the 63 μm – 2 mm fraction. Accession number D000.000.389.572.

**"Complex conodonts":** ***Wurmiella excavata.*** An M element of an ozarkodinin species collected from the Ludfordian (upper Silurian) of Gotland, sample no. G14-19OB (Supplementary Fig. 1D). The rock piece from which the conodont element was extracted was picked up by Oskar Bremer from loose beach blocks at Barshageudd 2, located at the southernmost tip of Gotland (Swedish grid coordinates WGS 84: 56°; 54′20.0″ N, 18°11′16.0″ E). The collected material were loose scree blocks and the collection site is located on public land, thus no permit was required. The blocks were dissolved in buffered acetic acid at the Department of Geology at Lund University, according to the technique of Jeppsson & Anehus (1995)[71]. The phosphatic remains were separated using heavy liquid separation. For details of age and the locality, see Bremer et al.[74]. Accession number D000.000.389.573.

**"Platform bearing complex conodonts":** ***Tripodellus gracilis.*** A $P_1$ element extracted from an upper Famennian sample collected at the Kowala Quarry in the Holy Cross Mountains (central Poland). The sample number Ko-338 was located just above the Kowala Shale, which forms a distinct marker horizon in the quarry. It was collected in a private quarry with permission. For details of age and the locality, see Ziegler and Sandberg[75]. The rock fragment was processed at the Institute of Paleobiology, Polish Academy of Sciences, in Warsaw: it was dissolved in 10% acetic acid, and the insoluble residue was enriched using an electromagnetic separator. The specimen (Supplementary Fig. 1E) is stored in the collections of the Institute of Paleobiology,

Polish Academy of Sciences, under the accession number ZPAL C16/3129.

**"Platform bearing complex conodonts": *Palmatolepis* sp.** A $P_1$ element of a palmatolepid species (Supplementary Fig. 1F) extracted from a limestone sample collected from the base of the Upper Kellwasser Horizon (Frasnian) at the now disused Schmidt Quarry located on public land outside of the hamlet of Braunau in the western part of the Rheinisches Schiefergebirge (central Germany). No permits are required for collecting and exporting common fossils from Germany[73]. The rock sample was processed at GeoZentrum Nordbayern (Erlangen, Germany): it was dissolved in 7% buffered acetic acid according to the method of Jeppsson & Anehus (1995)[71], wet sieved, and conodonts were picked from the 63 μm – 2 mm fraction. For details of age and the locality, see Sandberg et al.[76]. Accession number D000.000.389.575.

## Preparation and EBSD data acquisition

Electron backscatter diffraction (EBSD) is a scanning electron microscopy (SEM) based analytical technique that can be used to quantify crystal properties such as size, shape, and orientation in situ in direct comparison to the functional surface of the element. Owing to the bioapatite's high sensitivity to damage by the electron beam, previous studies faced beam damage to such an extent that crystal orientations could not be analyzed[47,77]. Thanks to improved sample preparation[78], the creation of an extremely flat surface, and access to highly sensitive CMOS-EBSD detectors[79], we were able to overcome this limitation. The samples were suspended in epoxy resin and polished using a succession of 6, 3 and 1 μm diamond suspension (for full protocol see Shirley et al.[78]). They were mounted on SEM stubs and coated with 3.5 nm of carbon coating. Backscatter electron images can be accessed in Supplementary Data 1 and source EBSD files are provided by Shirley et al.[80]. The initial data (*Bispathodus* cf. *aculeatus)* were collected using a Hitachi SU70 FEG SEM. *Pro. muelleri*, *Pan. equicostatus* and *W. excavata* EBSD data was acquired at the Department of Werkstoffwissenschaften WW1 of Friedrich-Alexander-Universität Erlangen-Nürnberg on a Helios NanoLab 600i DualBeam. *T. gracilis, Palmatolepis* sp. maps were collected at the Electron Microscopy Centre of Utrecht University using a Zeiss Gemini 450. All SEMs were fitted with an Oxford Instrument Symmetry EBSD detector using the AZtec software. This detector combines an optimized CMOS sensor with fiber optics, delivering a significant sensitivity increase compared to previous-generation EBSD detectors. This sensitivity, when combined with meticulous sample preparation, allowed us to lower the electron dose required for each diffraction pattern, thus minimizing tissue damage by the electron beam. Acquisition parameters were adjusted individually to maximize indexing rates (Supplementary Table 2). Indexing rates were between 57% and 87% for sample areas shown in Fig. 2, which is low compared to abiotic minerals, but very high for beam-sensitive materials. The dominant phase detected was apatite with lattice parameters listed in Supplementary Table 2 with an admixture of hydroxyapatite. Hydroxyapatite was converted to apatite for subsequent analysis.

## Raman spectrometry

All six sectioned conodonts were analyzed with Raman spectrometry using a WiTec alpha 300 Raman microscope equipped with a 532 nm laser and a grating of 1800 grooves per mm$^{-1}$. The system was calibrated using a white light prior to use, and instruments' spectral shifts were monitored using a silicon wafer standard. A 50× long working distance lens from Zeiss was used for the analysis, which provided a lateral spot size of 1 μm and approximately 2 μm depth penetration. The laser power was lowered to 2 mW to shield the samples from destructive effects of the Raman laser. Five Raman spectra were acquired per specimen beginning at the centre of each denticle moving towards the outer edge (i.e. occlusal surface) in 5 μm steps. Each spectrum was acquired for 10 seconds and averaged over 10 acquisitions to optimize the signal to noise ratio. Peak center at maximum intensity (PCMI) and full width at half maximum (FWHM) of the $v_1$-$PO_4^{3-}$ band in the Raman spectrum of apatite were calculated using a pseudo-Voigt function in Project FIVE 5.2 software by WiTec.

Raman spectra, band positions, and widths (PCMI and FWHM, respectively)[28] were extracted from the Project FIVE 5.2 software and plotted using R Software 4.3.0 (Fig. 4). The $v_1$-$PO_4^{3-}$ band parameters were then compared with those published in other studies[15,41,42]. Ranged major axis regression of FWHM over PCMI was calculated for each study using the package lmodel2[39,81].

## Quantification of crystal texture

EBSD data was processed in the MTEX 5.9.0 toolbox for MATLAB[82–84] using code provided by Shirley et al.[39]. Once imported, acquired areas were cropped to focus on the functional surfaces. The quality of the data obtained was visually assessed with band contrast images (Supplementary Fig. 4). No grain reconstruction, filtering or smoothing has been applied to data shown in Figs. 2, 3, or 7. Two maps of crystallographic orientations were constructed for each specimen. The X, Y, Z axes refer to directions in the specimen (Supplementary Fig. 5).

The X axis is defined by the long axis of the denticle. The Y axis points to the north and the Z axis into the plane of view. The colors of the maps represent the crystallographic orientations at each pixel of the map, relative to the X and Z axes, respectively. This is in contrast to the pole figures, which are also included in Fig. 2, which shows the distribution of crystallographic directions in the specimen coordinate system. The contoured upper hemisphere pole figures were created using the calcDensity function in MTEX using the la Vallee Poussin kernel and a halfwidth parameter of 4° (Fig. 2).

## Calculation of the Texture Index: equations

$$\text{ODF f}(\mathbf{g}) = \frac{\mathrm{d}V}{V \times \mathrm{d}g} \tag{1}$$

Orientation distribution function (ODF), a function on the orientation space that associates each orientation (**g**) to the volume percentage (*V*) of individual measurements in an area with this orientation[32]. *V* is the sample volume, d*V* is the volume of all grains *i* with the orientation *g* in the orientation range (angular element) $\mathrm{d}g = \frac{1}{8\pi^2}\sin\Phi\,\mathrm{d}\varphi_1\mathrm{d}\Phi\mathrm{d}\varphi_2$, $\oint \text{f}(\mathbf{g})\mathrm{d}g = 1$ and $\varphi_1, \Phi$ and $\varphi_2$ are Euler angles.

$$\text{TI} = \int \text{ODF f}(\mathbf{g})^2\mathrm{d}g \tag{2}$$

Texture Index, a measure of deviation of individual recorded orientations (*g*) from the orientation distribution function (ODF) calculated for the entire studied area. For ODF f(**g**) and d*g*, see Eq. 1.

## Statistical evaluation of the Texture Index between taxa

The null hypothesis that the TI values for all taxa originate from the same distribution was tested using the non-parametric Kruskal-Wallis test as implemented in R Software[85] using the code provided by Shirley et al.[39]. Pairwise tests between samples, reported in Supplementary Table 3, were carried out using the Wilcoxon test in the same software environment, with the Bonferroni correction applied. Confidence intervals reported in Supplementary Table 4 were calculated with the Bonferroni correction, i.e. the significance level of 5% was divided by 6 (number of samples), resulting in a significance level for each pairwise comparison equal to 0.83% and a corresponding confidence interval of 99.17%.

## The effect of measured area on Texture Index

Sizes of conodont elements and other vertebrate dental organs span multiple orders of magnitude, reflecting their range of body sizes and sizes of consumed prey. To investigate whether areas of functional surfaces might confound TI as a measure of the degree of structural control, we investigated the relationship between TI and the area from which it is calculated. For each sample (the entire scanned area, which might have been larger than the functional areas shown in Fig. 2), between 184 and 332 squares had been subsampled. For each square, ODF, TI, M-index[34] and pfJ[31] were calculated. To test the null hypothesis of no relationship ($\rho = 0$) we calculated Pearson's correlation coefficient ($\rho$) between TI and the area. As comparison, well-documented biominerals, EBSD datasets of aragonitic nautiloid nacre and oyster prismatic layer were subject to the same procedure (Supplementary Table 1). All tests show a negative correlation between the TI and area, significant at $\alpha = 0.05$. It shows that the larger an examined area is, the lower the TI will be (Supplementary Fig. 3c). This indicates that the size of larger specimens, which in our dataset tend to have more complex morphology (i.e. *Bispathodus* and *Wurmiella* are larger than the coniform taxa), does not explain higher TI values.

## Comparison of metrics for quantifying crystal texture

There are several methods to quantify crystal texture from a material, the majority of which are derived from the Orientation Distribution Function (ODF), a mathematical representation of the distribution of crystallographic orientations within a polycrystalline material. TI and M-index are both measures of the degree of crystallographic texture in a material. However, they are calculated using different methods and are based on different mathematical representations of the texture.

Texture Index (TI, also referred to as the J-index[33]) quantifies the maximum intensity of the (ODF), providing information about the dominant orientation(s) in the material. TI is commonly applied to geological materials to measure fabric strength[34]. This metric, however, has the disadvantage of requiring a substantial number of discrete orientations, which make it impractical to obtain a meaningful measure of fabric strength and its difficulty to interpret[86]. The M-index, on the other hand, characterizes the spread or distribution of crystallographic orientations, indicating how broad or diffuse the texture is (Supplementary Fig. 4).

This has the potential to provide more meaningful information on the crystal texture, in particular with datasets with limited orientation data. Another method is Pole Figure T-Index (*pfJ*), a measure of crystallographic texture in a material utilizing the same method as the TI, specifically based on data obtained from pole figures. This allows crystal texture to be quantified relative to an individual crystal axis.

We conducted texture analysis on our six samples using a combination of TI and M-Index measurements (Supplementary Fig. 4). TI exhibited a consistent trend across all samples, showing a clear increase in values indicative of a stronger and more pronounced texture with evolution. This suggests a progressive shift towards a preferred orientation within the denticle tissue. The M-Index showed a similar increase, but it was less pronounced compared to the changes observed in TI. Several factors may contribute to this difference. One possibility is that TI is sensitive to textures with high maximum intensities. The M-Index, which predominantly characterizes the spread of orientations, may show a more moderate response to changes in maximum intensity. A similar trend was seen in the pfJ for the *a*-axis and *c*-axis, where the differences are smaller than in the M-Index. It appears from our results that, in the context of textures analyzed here, TI has more sensitivity to subtle changes to texture differences, allowing for a higher degree of separation between samples.

We tested the influence of the sample area on these texture indices. We found that TI exhibited a consistent decrease as the area of measurement increased. This trend suggests that TI is sensitive to the measured area. In contrast, the M-index presented a more stable response to changes in the sampled area. It maintained a relatively constant value in the same material regardless of the spatial extent. This may indicate that the M-index may be less influenced by variations in the area of measurement (Supplementary Fig. 4).

Both indices investigated appear to support our hypothesis on increasing crystallographic organization. Here we predominately discuss TI due to its apparent higher sensitivity to more subtle changes in crystallographic texture (Supplementary Fig. 4).

## Reconstruction of crystal grains

Conodont skeletal tissues form grains, also referred to as crystallites, which are visible in topography in broken or etched sections[13,14] and in EBSD maps (Fig. 2). In EBSD datasets, algorithms based on misorientation angles are applied to calculate grain sizes. This allows inferring properties of the grains such as the area, grain orientation and length. However, grain reconstruction is done based on a threshold misorientation angle which, in most cases, is obtained from other analytical methods. For enamel and enamel-like tissues such as conodont hyaline tissue, to the best of our knowledge, no such threshold has been established. We carried out multiple grain reconstructions for conodont crown tissues using threshold values from 1° to 17° with a step of 4°. Supplementary Fig. 2 illustrates an example of calculation of grain boundaries compared to a band contrast map of a sample subset of *Bispathodus* cf. *aculeatus*. Band contrast is a measure of image quality derived from the Hough transformation. It describes the average intensity of Kikuchi bands in comparison to the overall intensity within the area of interest. The example in Supplementary Fig. 2 shows an end-product of grain reconstruction using the `calc-Grains()` function of MTEX set to follow non convex outer boundaries based on a user defined misorientation angle. After the first round of grain reconstruction, grains smaller than 2 pixels were discarded and the function was applied again to eliminate spurious grains appearing as a result of misindexed pixels. Following this the data was then filled in Supplementary Fig. 2 using the `fill()` function of MTEX which attempts to fill in grains with pixels of similar orientations. The resulting maps (Supplementary Fig. 2) may convey an impression of well resolved texture with a great variety of grain sizes and shapes. However, comparison with the band contrast image reveals that the inferred grain boundaries mostly correspond to non-indexed areas. Thus, obtained grain distribution would reflect the pattern of missing data and not the "true" grain shapes and sizes. No cleaning algorithms were used in our post processing workflow. We identify two major barriers to grain reconstruction in conodont tissues: (1) low indexing rate due to beam damage and challenging preparation, (2) internal heterogeneity of tissues, such as small misorientation angles between adjacent grains, typical for biominerals and often reflect the mechanism of formation.

## Reporting summary

Further information on research design is available in the Nature Portfolio Reporting Summary linked to this article.

# Data availability

The datasets generated during and/or analyzed during the current study are available in the following repository entry: Shirley et al., 2024. Supplementary Data and Code for Increasing Control over Biomineralization in Conodont Evolution. Open Science Framework. https://doi.org/10.17605/OSF.IO/C2NU3. Source Data for Fig. 2 can be found at https://osf.io/m26qa/; for Figs. 3 and 4 at https://osf.io/yfk3x; for Fig. 5a at https://osf.io/smhjr and for Fig. 5b at https://osf.io/cgexs; for Fig. 6 at https://osf.io/h85k4; for Fig. 7 at https://osf.io/vnq6s/. No external data was used to produce Fig. 1. Supplementary Information is available for this paper. Correspondence and requests for materials should be addressed to Emilia Jarochowska.

## Code availability

Code used in the manuscript is available as: Shirley, B., Leonhard, I., Murdock, D., Repetski, J., Świś, P., Bestmann, M., Trimby, P., Ohl, M., Plümper, O., King, H., & Jarochowska, E. (2024). Code for "Increasing control or biomineralization in conodont evolution" (v1.0). Zenodo. https://doi.org/10.5281/zenodo.11222479.

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

## Acknowledgements

BS and EJ were supported by Deutsche Forschungsgemeinschaft (project JA 2718/3-1). BS was also supported by the EXCITE Network and DAAD. We thank B. Leipner-Mata for help in sample preparation and J.D. Loch and J.F. Taylor for collecting the sample with Cambrian conodonts. This publication results from work carried out under Trans-National Access action under the support of EXCITE - EC- HORIZON 2020 -INFRAIA 2020 Integrating Activities for Starting Communities under grant agreement N.101005611.

## Author contributions

**Bryan Shirley** Conceptualization, Data curation, Formal analysis, Funding acquisition, Laboratory work, Data collection, Software development, Visualization, Writing. **Isabella Leonhard** Laboratory work, Data collection. **Duncan Murdock** Provision of samples, Writing. **John Repetski** Provision of samples. **Przemysław Świś** Provision of samples, Laboratory work. **Michel Bestmann** Development of methodology, Validation, Writing. **Pat Trimby** Development of methodology, Data collection. **Markus Ohl** Development of methodology, Data collection. **Oliver Plümper** Writing, Funding acquisition. **Helen King** Laboratory work, Data collection, Development of methodology, Writing. **Emilia Jarochowska** Conceptualization, Data curation, Development of methodology, Formal analysis, Provision of samples, Writing, Funding acquisition.

## Competing interests

The authors declare no competing interests.
