## [Peer Review File · Nature Communications]

Increasing control over biomineralization in conodont evolutionReviewers' Comments:

Reviewer #1:

Remarks to the Author:

This is a very interesting manuscript and I concur that, if and once revised appropriately, it may be published in Nature Communications.

The first vertebrate skeleton is found in conodonts as dental elements made of apatite. Thanks to the first successful application of Electron Backscatter Diffraction to bioapatite, the authors show that the crystallographic order varies from one sample to another. They argue that the crystallographic order increases throughout the evolution of conodont teeth, in parallel with morphological adaptation to food processing, thereby suggesting increasing control over biomineralization throughout the evolution of the first skeletonized vertebrates. This opens interesting perspectives, also for the application of this technique to other vertebrates.

Yet, in my opinion, the authors fail, for now, to justify convincingly their sampling choices: they analyzed 6 taxa only and they argue they are representative of 'increasing degrees of adaptation to dental function'. Besides the fact that I do not understand what 'adaptation to dental function' exactly means and how this was quantified here, there seems to be a risk of circular reasoning and/or of 'cherry picking' the data that agrees with one's hypothesis. To me the authors' figure 2a does not necessarily demonstrate a temporal trend of texture index but rather suggests that texture index increases from coniform to blade to platform conodont elements. Since these three types of conodont elements can also co-occur, the authors should discuss the polarity of this sequence in conodont evolution (btw, does it really reflect increased 'sophistication?'). I would also encourage them to discuss the possibility that these three geometries correspond to three (or more) ways to process food and whether their hypotheses about the occlusal axes are robust and whether considering only uniaxial compressive stresses is relevant here.

I am looking forward to reading the revised version of this ms.

Best regards,

Nicolas Goudemand

other (mostly minor) comments:

line 24: 'throughout the evolution of conodont teeth': not convinced that these 6 taxa represent the whole story. Please nuance.

line 57: 'precise occlusal': 'precise' does not seem to be the right term here

line 65: 'digestion': replace with 'processing'?

line 71: 'for many conodont lineages': replace 'many' with 'some' ?

line 84: 'load': only when that load is uniaxial compression

line 93: precise what is 'V'

line 104: what do you mean with 'increasing degrees of adaptation to dental function'? Coniform elements may be as adapted to their particular function as blade or platform elements are to theirs...

line 114: how do you define those axes? What if the mastication movement includes rotations?

line 122: 'orientations'

lines 120 and 124: what are those brackets?

lines 146: what if one analyzes a more 'derived' (e.g. Triassic) coniform element and a less 'derived' platform element?

line 175: 'least specialized feeding in Proconodontus' : How do you know that feeding in Proconodontus is less specialized than in Palmatolepis?

line 199: 'however': ?? unclear logical relationship

line 212: 'sophistication': not demonstrated

line 233: too strong a statement

line 236: 'selectively': selectivity?

Reviewer #2:

Remarks to the Author:

This is a significant and nicely illustrated conodont manuscript describing the variation of degree of crystallographic order in selected conodont material. The approach is innovative. My main criticism is that expanding conclusions to all groups of early vertebrates is hazardous.

In addition, the introductory part is overestimating the role of vertebrates.

- Line 17: "outstanding diversity of skeletal tissue": what about calcareous shelled skeletal organisms? They did the same (also in the microworld) or even better.

- Line 18: "extreme hardness and elasticity": siliceous shells are even harder. Furthermore, elasticity does not reflect simple mineralization, but strongly depends on the presence/quantity of organic material.

- Line 20: "teeth" or "tooth-like" elements as reported in line 52?

- Lines 35-37: "Vertebrates are ... the one that has achieved the highest diversity and disparity": What about insects? As regards disparity, it is a delicate topic: disparity is not only morphology, but disparity can be any phenotypic aspect including behaviour and ecology.

- Line 45: "first vertebrates": the manuscript deals only with conodonts (six specimens ranging in age from the Cambrian to the Devonian). In that time frame many other vertebrates appeared in the sea and were going to jump on the land. Why wasn't that material investigated?

In general:

- What about the role of diagenesis on your material?

- You base your conclusions on a very few specimens (six): how confident can you be on such a limited set?

Reviewer #3:

Remarks to the Author:

1. Short summary of opinion

This MS was of interest because EBSD analysis from apatite biominerals are very rare and is in its infancy, so this study has a good potential to be a significant study. At the same time, unfortunately, the robustness of data ("sufficiently strong evidence") is not as good as what I expect for a usual Nat. Comm paper.

2. Key results

Shirley et al. analyzed six conodont taxa (from "primitive" to "derived" ones) with EBSD, which is a superb tool for crystallographic data of (bio)minerals. The authors succeeded in getting the orientation maps and pole figures, which were interpreted in the context of conodont evolution. To my knowledge, earlier studies tried EBSD analysis to conodont, and some were even successful at getting a good map but getting the data from several taxa is the major accomplishment of this study. I myself once tried EBSD analysis to bones (bioapatite) and was disappointed by the absence of any Kikuchi bands, so I fully understand that what the authors accomplished herein is a very good contribution.

3. Validity

With that being said, I have concerns for the validity of the research. Please note that I do not mean that it is not a good study design. I just considered that Nat. Comm. is a very selective journal, so I also used stringent criteria, which I expect for a paper published in Nat. Comm.

3.1 Here, the authors did not consider any taphonomic status of each specimen. I felt that the authors treated the specimens as if they were neontological specimens, which are free from taphonomic consideration. For example, *Proconodontus muelleri* is Cambrian in age, and *Palmatolepis* sp. is Devonian in age, so there is nearly 100 Myr time gap between the two specimens. So one may expect that 'biogenic' minerals in Cambrian *Proconodontus muelleri* is comparatively less well-preserved than those of Devonian *Palmatolepis* sp. and it may influence the difference between the crystallographic

alignments of two specimens. But there is no consideration like this in the current study. Besides, even if ages of two specimen are similar (e.g. Devonian Tripodellus and Palmatolepis), spatial setting can affect the preservation. For example, when a formation of conodont A was deeply buried or under the influence of significant tectonic pressures and heat, whereas conodont B was buried in a 'peaceful formation', which was relatively free from pressure and heat, the preservation status of conodont A and B may not be the same.

In my opinion, neglecting the taphonomic status of specimens can significantly undermine the validity of this study. Please note that consideration of taphonomic status is an important topic for many paleontological studies (e.g., see a great example of Casella et al. 2018, *Palaeo3*, <https://doi.org/10.1016/j.palaeo.2018.03.011>).

3.2 Although I fully understand getting EBSD maps from conodonts is the very hard task, I still have concern for the small sample size (i.e., 6). There are many families in conodont (<https://en.wikipedia.org/wiki/Conodont>), so I expect that the number of genera is larger than that of families. So 'six' may be an insufficient number to represent the pattern in conodont evolution and it may weaken the robustness of the conclusion. In other words, a future study that uses much more diverse conodont taxa may report a different conclusion, and the current study is vulnerable to this type of challenge. In addition, if I understand correctly, just a single map was acquired from each specimen? In my experience, the crystallographic orientation (i.e. pole figure) is pretty local (especially a-axis direction), so if a different part of the same specimen is analyzed with EBSD, the results (e.g. pole figure) may be different. In other words, just a single map per a specimen may not guarantee the solid reproducibility, which is required for Nat. Comm. papers. Again, please do not get me wrong. I truly think 'six' is already a great accomplishment, but I just feel that it is somewhat weaker than the 'level of support' criterion of Nat. Comm.

4. Significance

When it comes to significance, I think it's A+. Because this research proved that spatiotemporally diverse conodonts can be analyzed with EBSD, many conodont researchers will adopt EBSD for their arsenal. In addition, many other vertebrates have teeth (e.g. fish, Felidae, T. rex, the authors, the reviewer [though some are replaced with amalgam], etc), so the accomplishments of this study will influence many fields of science such as paleontology, odontology, zoology, etc.

I just wish that the contribution of this study and that of Atakul-Özdemir et al. (2021, R. Soc. Open Sci.) should have been clearly explained in the main text to guarantee the significance of this study.

5. Data and methodology

5.1 About the methodology: The panel B of Extended Data Figure 2 used a 'cleaning' to enhance the mapping. However, compared to the panel A, the cleaned maps in the panel B look 'too clear' and it may mean that the authors used too radical cleaning method for the panel B.

If this cleaning method was used in Figure 1, I would say that Figure 1 may contain pretty much 'false-positive' data, which are attributable to the radical cleaning. If Figure 1 used 'raw data', which was not influenced by any cleaning, please dismiss my current comment (and I believe the authors did not apply 'radical cleaning' to Figure 1).

Anyway, please provide this sort of information and all other steps applied to present Figure 1 to guarantee the full reproducibility. Currently, I felt that the method section is not detailed enough, so some steps could not be evaluated with enough confidence.

5.2 About the quality of presentation: I think there are some room for improvements. Please see below "11. Additional comments".

6. Analytical approach

This issue was explained in the '3' and '5' above.

7. Suggested improvements

This part is directly related to '3. Validity' above.

7.1 Taphonomic consideration: I strongly recommend that Casella et al. (2018) be a helpful literature that explains why considering taphonomic status is important for biomineral studies. There, one can see that taphonomic influences degrade the calcite in brachiopods and it can make 'fake news' from the old past. Admittedly, calcite is soft mineral as authors pointed out, so bioapatite fossils like conodont may not be severely affected by diagenesis (against my concerns). However, at least investigating the taphonomic status of the materials used in this study may be able to defend the concerns of mine and potential future readers.

I suggest that investigating the Raman spectra of conodont may yield useful information at least for thermal alteration. Please see McMillan and Golding (2019, Palaeo3) for details. If it turns out that Cambrian materials are heavily thermally influenced while younger Devonian materials (the ones used in Figure 1) are less affected by thermal alteration, I think this factor may contribute to 'false-positive' pattern (highly aligned Devonian signals vs weakly aligned Cambrian signals) in Figure 1. So I wish that the authors eliminate this sort of possibility completely with additional Raman data.

At the same time, I know Raman spectroscopy is not always available, and learning new skills used in McMillan and Golding (2019) can be a harsh request as a revision. So I think somewhat more practical suggestion (though I prefer Raman data) would be to investigate the classic conodont Color Alteration Index of the materials used in this study. If CAI shows that older materials are not always more heavily thermally altered, I believe the pattern reported in this study can be more safely understood as the biological signal as authors claimed (against my "devil's advocate" concern).

7.2 Ideally, I believe more wider sampling of taxa or getting a few more EBSD maps from each taxon (to guarantee the reproducibility of Figure 1) will definitely improve the robustness of the conclusion. But I know how hard it is to prepare new EBSD specimens and running EBSD analysis. So I will leave it to the editor's discretion.

8. Clarity and context

The text is generally clear and explanation of context at Introduction is excellent (another A+). But some parts are hard to follow. Please see my further comments in "11. Additional comments". Also, I believe the title tries to encompass too wider implications. In my opinion, even if the title is simply constrained to conodont instead of wider 'early vertebrate', this study is equally awesome, so please consider this issue during the revision.

9. References

I am not a conodont specialist, so I am not qualified to officially say something about the quality of bibliography. That said, many great journals and renowned paleontologists' name in the bibliography made me believe that the literature cited are appropriate.

10. My expertise

When it comes to statistical analysis, I am not good at it, so please follow the opinion of other reviewers. About the validity of textural analysis and technical issues of EBSD analysis, such as Orientation Distribution Function and Texture Index, other reviewer(s) will provide more solid opinions than I can.

Also, because I am not a conodont expertise, I am not qualified to provide critical comments to Discussion. I hope other conodont experts can do this.

11. Additional comments (about writing)

11.1 Title

In my opinion, it needs to be more focused on the core contents. When I read the title, I anticipated that this study used wide variety of vertebrates fossils (especially diverse Agnatha of Paleozoic) or at least analyzed much more diverse taxa of conodont. Please do not get me wrong. I strongly believe that getting the crystallographic data from conodonts is a very important contribution, but current title may not be representative for the contents of the paper because all results were obtained from the conodonts, and someone may think that just one clade (=conodont) is not enough to represent 'early vertebrate'. Although I know that a concise title is preferred for a journal like Nat. Comm., I still believe that the best title should be a direct indicator for the contents.

11.2 Abstract

It is a well written summary. I point out two things: 1. There are four citations, but to my knowledge, Nat. Comm. does not allow citations in abstracts; 2. It is composed of more than 150 words, which is a length limit for abstracts of Nat. Comm.

I feel the current abstract is more like a that of Nature, so please revise the abstract to meet the guideline of Nat. Comm.

11.3 Introduction

The background information is very well presented and justifies the research. However, the arrangement needs improvements. The findings of other research and the contribution of current research is somewhat jumbled. I feel clear separation of 'what others did (i.e., literature review)' and 'what we did in this research (i.e., Here we, blah blah)' will make a better Introduction.

The authors also mentioned c-axis of apatite and it is an essential terminology for the study. My small concern is that very few readers will know what 'c-axis' means. I would recommend that the authors prepare a simple supplementary figure to show a unit cell apatite and its axes direction. I believe it will help readers to follow your thoughts.

Please see also my minor comments in the annotated PDF.

11.4 Methods (Preparation and data acquisition)

If space permits, please briefly explain how this study succeeded in getting clear Kikuchi bands from the polished surface of conodont unlike earlier research, which were not that successful. Although Shirley et al.¹⁰ was cited, readers may want to know the difference without visiting Shirley et al.¹⁰. Please see also my minor comments in the annotated PDF.

11.4 Extended Data

(1) Figure 1. Although SEM images are very good at showing the morphological details, I recommend stereoscopic microscope images (e.g. Panels D, E) because they show color. I think color image is especially important (please see my "7.1 Taphonomic consideration" above). If possible, please provide.

(2) Figure 2(a). It is very hard to find any difference among the different grain boundary thresholds. It is understandable because just one degree of difference is very subtle, but I am not sure then what angle is the most suitable for the match between 'human eye' and grain boundary threshold. I anticipated the answer from the authors somewhere in Lines 441–454, but there was no answer (also in Lines 135–138). It may confuse readers.

Figure 2(b). Because there is no detailed information for the "standard grain smoothing and filter (here half-quadratic)" (Line 446), I cannot provide clear feedback. However, if 'Filled grains' maps are the results of 'cleaning (=extrapolation of pixels)' of the original data presented in Panel (a), I think it is problematic because the 'cleaning' step was too aggressive and radical. As authors pointed out, the 'boundaries' in the cleaned maps may not show the precise positions of real boundaries, grain shapes, and sizes. Most importantly, orientation information may be affected by this aggressive, anthropogenic factor. Did the authors use the "standard grain smoothing and filter (here half-quadratic)" cleaning method to make Figure 1 (I hope not)?

11.6 Results

(1) Lines 118–122: Here, the description is very hard to follow. What are the 'elongated apatite grains'? There are many shapes of grain in SIX EBSD maps, so it's hard to know what I should see. In addition, due to the diverse shapes of apatite, I am pretty skeptical that the description is generally applicable to the maps in Figure 1.

(2) Figure 1: Although the maps are most probably 'inverse pole figure maps', no legend was provided so it is hard to fully understand the meaning of colors in the maps. In addition, there is no explanation for the numbers above the pole figures. I believe they are MUD (multiple of uniform density) and if it is correct, please explain at the caption.

11.7 Discussion

(1) Lines 197–202: Similar to Lines 118–122, the authors keep mentioning 'elongated grains', but it is not that clear to readers. Please clarify what are 'elongated grains' and get more generality for this observation.

(2) Lines 218–223: Perhaps due to the too long sentence, I cannot clearly understand what this part tries to explain. Could you please rephrase?

Reviewer #4:

Remarks to the Author:

Review to the manuscript titled "Increasing control over biomineralization in early vertebrates".

The work documented by Shirley et al. suggested the evolution of the first mineralized vertebrate tissues in conodont by progressive adaptation to dental function. The indexing of crystallographic axes of apatite in conodont fossil by EBSD analysis was successful and the results well showed that c-axes of apatite consisting of hypermineralized crown tissues were aligned parallel/subparallel to the occlusal (biting) axes. It is interesting that the preferred crystallographic orientations of conodont's crown tissues could be developed by evolving conodont feeding mechanisms. However, I feel that further consideration of two issues is required to interpret the EBSD results.

1. How did you rule out the influence of diagenesis on the development of preferred crystallographic orientations of conodont fossils? You mentioned that "the hyaline tissue is widely used by geochemists thanks to its outstanding resistance to diagenesis (lines 67-68)", but this does not guarantee the resistance of recrystallization. The related reference (15. Zhang et al., 2017) also does not suggest that recrystallization of conodont apatite does not occur easily by diagenesis, but suggests the integrated approach to recognize the nature and degree of diagenetic alteration in conodont specimen. Is there any other evidence that development of preferred crystallographic orientations of biomineral did not be affected by diagenesis? In this study, the spatial distribution of conodont samples is varied, such as U.S.A., Ukraine, Germany, Sweden, Canada, and Belgium (maybe?). The time variation is also large, from upper Cambrian to upper Devonian. These may indicate that the environments of diagenesis of samples should be different. That is, how did you rule out the effect of pressure and temperature on the crystallographic texture?

In addition, you mentioned that "the discordance between the long axes of grains and the c-axes is unusual in abiotic minerals but common in biological tissues (line 121-122)". What are the long axes of elongated apatite grains? From what I understand, the long axis of apatite is the direction of maximum length of same colored grain in grain color maps within Figure 1. In Fig.1f, the elongated long axes of apatite are likely to be perpendicular to the crystallographic c-axis of apatite. However, this trend is rarely observed in other samples. In Fig.1a and 1b, the crown appears to consist of almost a large single crystalline grain. Especially, in Fig.1d, the elongated long axes of apatite are parallel to the c-axes of apatite. In this case, is there a possibility of recrystallization in these samples?

2. In this study, six hyaline tissues of conodont dental elements were analyzed for six taxa of conodonts. That is, only one section of hyaline tissue was selected for each taxon. In this case, each specimen was understood to be representative of each taxon of conodonts. However, is the cross-section representative sufficient to interpret the skeletal evolution of conodonts according to changes in feeding function? Of course, this result is the first successful analysis of EBSD for bioapatite, but I think that only one section for each taxon would not be enough to representative of one taxon of conodonts. Although *Pro. muelleri* and *Pan. equicostatus* have a single coniform element in each, the other four samples appear to have multiple hyaline tissues. Are there similar trends in EBSD results for other hyaline tissues in each specimen?

In addition to these major points, I'd like to suggest some recommendations for minor issues.

1. In Fig.1, what is the meaning of the reference frame (X- and Y-directions)? Do the black areas in schematic diagrams indicate the grain color maps in each specimen? From what I understand, you want to show the crystallographic c-axes of apatite grains aligned perpendicular to the occlusal surfaces (that is, aligned parallel to the occlusal (biting) axes). For this purpose, it is recommended to use an inverse pole figure (IPF) map in terms of the occlusal axis.

Of course, Fig.1 can show the c-axes aligned parallel to the occlusal surface, but an IPF or IPF map is generally used to show where the crystallographic axes of minerals are aligned in a specific direction (like in the reference 26. Kilian and Heilbronner, 2017). Because you processed the EBSD data using MTEX, a MATLAB toolbox, I believe that you can rotate the Euler angles to the specific direction. If X- and Y-directions are the sample coordinates in the SEM, you can make the X-direction aligned parallel to the occlusal surface, and Y-direction aligned perpendicular to the occlusal surface (that is, parallel to the occlusal axes). Then, when you plot the IPF map or IPF in terms of the Y-direction, it is easy to see which axis is perpendicular to the occlusal surface (in this case, maybe c-axis).

Otherwise, please indicate the meaning of the X- and Y- directions and match the directions between the schematic diagram and grain color map in each specimen.

2. You calculated the texture index (TI) to quantify the spread of crystallographic orientation of biogenic apatite. You tried many re-calculations of ODF and TI for random subsets to eliminate the bias of the analyzed size of sample, but you excluded the calculation of grain size because of no existence of known misorientation angle to discernible grain boundary in enamel and enamel-like tissues. Of course, it is not easy to define the reference angle of grain boundary. However, in materials science (generally used for alloys), boundary with misorientation angle of $\sim 15^\circ$ is used to define a high-angle grain boundary (indicating the general grain boundary), and boundary with misorientation angle less than 15° is considered as a low-angle grain boundary (indicating the subgrain boundary). In the field of geology, this angle is considered to be about 10° in the rock sample. In my opinion, the reference angle of the grain boundary for biogenic apatite would be similar to that of abiogenic (abiotic) minerals because of the similar mechanical properties between biogenic and abiogenic minerals. If you calculate the grain boundary of the sample, you can easily interpret the trend of TI values considering a grain size distribution.

TI values of *T. gracilis* and *Palmatolepis sp.* are higher than those of *Pro. muelleri* and *Pan. equicostatus* because former two samples are composed almost large single crystalline texture. The narrow TI value ranges of *Pro. muelleri* and *Pan. equicostatus* indicate the homogeneous grain size distribution. I don't know why grain size distribution is differ among the samples, but the crystallite of hyaline tissue appears to generally tend to become larger towards the later taxa of conodonts.

In addition, it seems that boundaries with misorientation angle less than 10° should be regarded as low-angle grain boundaries (that is, subgrain boundary). In that case, the kernel angle misorientation (KAM) map or grain reference orientation deviation (GROD) angle map is more suitable to show low-angle grain boundaries in the biogenic apatite minerals. The KAM map shows the misorientation less than the predefined threshold value calculated by the mean orientation between a point and its neighbors. The GROD map is generated based on the deviation between the mean orientation of a reference point and those of other points. You mentioned that "the misorientations vary between areas within one tissue depending on their position with respect to functional directions and the compressional and shear stresses acting in these directions (lines 133-135)". So, these maps can easily represent the intracrystalline misorientation within one tissue regardless of the existence of known misorientation angle to discernible grain boundary. It may require some data processing, but I believe that you can plot these maps by using codes within MTEX.

3. In lines 93-95, are reference numbers 28 and 29 correct for ODF? Please check and correct with

appropriate references (maybe Bunge, H.-J., 1982. *Texture Analysis in Materials Science: Mathematical Models*. Butterworths, London, UK).

In lines 100-101, a texture index (TI) also needs a citation. Please cite the appropriate references (probably Bunge (1982), or generally used as "J-index" by Mainprice and Silver (1993)).

If you want to just quantify the sharpness of pole figure of c-axes, pole figure J-index (pfJ-index) is also considerable. This value is analytically defined in a similar manner of J-index (Michibayashi and Mainprice, 2004), but as far as I know, there are no code in MTEX unfortunately.

If you want to quantify the intracrystalline misorientation within a hyaline tissue, it is considerable to use a "misorientation index (M-index)" calculated from the misorientation angles among the uncorrelated grain pairs (Skemer et al., 2005). Unlike J-index (texture index), M-index has an advantage that it is hardly affected by the number of data, but also has a disadvantage of loss of spatial and textural information of the original dataset. I'm not sure whether you can get more reasonable result than J-index, but it is worth considering the M-index to quantify the crystallinity of hyaline tissue. However, be careful if you calculate the M-index by MTEX code. You have to do it directly using raw EBSD data, not using the result of ODF calculation.

Overall, the results of this manuscript will be interesting to geologists, especially mineralogist, and I think it may have the potential to provide important inspiration to paleontologists. However, several issues require clarification before it is considered by Nature Communications, at least for major issues.

Response to comments by the reviewers

Reponses are marked in blue

Reviewer #1 (Remarks to the Author):

This is a very interesting manuscript and I concur that, if and once revised appropriately, it may be published in Nature Communications.

The first vertebrate skeleton is found in conodonts as dental elements made of apatite. Thanks to the first successful application of Electron Backscatter Diffraction to bioapatite, the authors show that the crystallographic order varies from one sample to another. They argue that the crystallographic order increases throughout the evolution of conodont teeth, in parallel with morphological adaptation to food processing, thereby suggesting increasing control over biomineralization throughout the evolution of the first skeletonized vertebrates. This opens interesting perspectives, also for the application of this technique to other vertebrates.

Yet, in my opinion, the authors fail, for now, to justify convincingly their sampling choices: they analyzed 6 taxa only and they argue they are representative of 'increasing degrees of adaptation to dental function'.

The number of specimens reflects the difficulties of this analytical methods. The six taxa is the outcome of a 5-year project. This is a second study in which any EBSD data has been obtained and the only previous one (which corresponded to an entire postdoctoral project) examined only one taxon and yielded data of limited use, because the datasets and processing methods were not provided. In our sampling, we focused on the earliest and on the least derived taxa to capture the early stages of the evolution of biomineralization. Therefore the youngest taxa are represented poorly: they show smaller disparity than the early conodonts, based on which we made the assumption that the highest ultrastructural diversity should be sought in older taxa. The complete datasets and code are provided with the manuscript, which is unique for a EBSD study in palaeobiology.

Besides the fact that I do not understand what 'adaptation to dental function' exactly means and how this was quantified here, there seems to be a risk of circular reasoning and/or of 'cherry picking' the data that agrees with one's hypothesis.

The "adaptation to dental function" in this study follows the definition proposed by Donoghue (2001), as we directly test his hypothesis. This has been spelled out in the text. Furthermore, we specify that "adaptation to dental function" is not a general metric measured the same way in all organisms. It is related to how the tooth (or its individual surfaces) operates, e.g. if it's a shearing, crushing or grinding tooth and in which directions it occludes. We specify in the text that in case of conodont denticles or single cones, that adaptation is reflected in the strongest resistance to uniaxial compressional stress.

To me the authors' figure 2a does not necessarily demonstrate a temporal trend of texture index but rather suggests that texture index increases from coniform to blade to platform conodont elements. Since these three types of conodont elements can also co-occur, the authors should discuss the polarity of this sequence in conodont evolution (btw, does it really reflect increased 'sophistication'?).

This seems to be a misunderstanding between a temporal succession and the degree of evolutionary change. A younger taxon does not need to be more derived than an older one. "Primitive" (preserving many ancestral characters) organisms can coexist for long periods with their more derived relatives. Therefore we do not present a temporal trend and in that sense the reviewer is correct that this is not what Figure 3a (formerly 2a) shows. It shows a phylogenetic tree with **evolutionary distance** distinguishing the taxa. Derived conodonts are those showing more specialized morphologies. Thus, more complex and more derived morphologies can co-occur with simpler and less derived ones: they are still separated by a long evolutionary distance.

I would also encourage them to discuss the possibility that these three geometries correspond to three (or more) ways to process food and whether their hypotheses about the occlusal axes are robust and whether considering only uniaxial compressive stresses is relevant here.

We distinguished between occlusion models of entire apparatuses and the occlusal axes of single denticles and cusps. The distinction is now emphasized in the text. The article focuses on comparison of the single morphological unit (denticle or cusp) in a conodont element, which can be compared across the entire lineage.

line 114: how do you define those axes? What if the mastication movement includes rotations?

We do consider rotations and the asymmetry of the movement in the results and discussion. But we do not go into this detail in the introduction because it will be confusing for the reader before they see the figures.

All the corrections below have been applied:

other (mostly minor) comments:

line 24: 'throughout the evolution of conodont teeth': not convinced that these 6 taxa represent the whole story. Please nuance.

line 57: 'precise occlusal': 'precise' does not seem to be the right term here

line 71: 'for many conodont lineages': replace 'many' with 'some' ?

line 84: 'load': only when that load is uniaxial compression

line 93: precise what is 'V'

line 104: what do you mean with 'increasing degrees of adaptation to dental function'? Coniform elements may be as adapted to their particular function as blade or platform elements are to theirs...

line 122: 'orientations'

line 175: 'least specialized feeding in Proconodontus' : How do you know that feeding in Proconodontus is less specialized than in Palmatolepis?

line 199: 'however': ?? unclear logical relationship

line 212: 'sophistication': not demonstrated

line 233: too strong a statement

lines 120 and 124: what are those brackets?

- These brackets describe a set of directions or axes equivalent to [hkl] under the symmetry group of the crystal; a direction or axis in the set. This is standard convention in mineralogy; we did not see the point of explaining it in the main text since it is not required to follow the text for a person without a mineralogical background and anyone familiar with mineralogy will be familiar with this notation.

lines 146: what if one analyzes a more 'derived' (e.g. Triassic) coniform element and a less 'derived' platform element?

Younger does not mean more derived, as explained earlier.

line 65: 'digestion': replace with 'processing'?

"Digestion" is commonly used in the literature in this context so we decided not to follow this suggestion.

Reviewer #2 (Remarks to the Author):

This is a significant and nicely illustrated conodont manuscript describing the variation of degree of crystallographic order in selected conodont material. The approach is innovative. My main criticism is that expanding conclusions to all groups of early vertebrates is hazardous.

We revised the text to assure that our conclusions focus on conodonts. Those mentions of vertebrates which explain the broader context of the study were left in the text. Although we focused on conodonts as the earliest biomineralizing vertebrates, our study proposes a methodology that applies to other vertebrate groups.

- Line 17: "outstanding diversity of skeletal tissue": what about calcareous shelled skeletal organisms? They did the same (also in the microworld) or even better.

That's across the many calcifying phyla. Any single phylum of calcifying organisms will show a much smaller skeletal diversity. The sentence emphasizes diversity within one taxonomic group (vertebrates), not a competition between organisms with phosphatic vs. calcareous skeletons.

- Line 18: "extreme hardness and elasticity": siliceous shells are even harder. Furthermore, elasticity does not reflect simple mineralization, but strongly depends on the presence/quantity of organic material.

We agree with the reviewer. The sentence has been corrected to "outstanding combination of hardness and elasticity", which is unique to vertebrate tissues. Siliceous shells are harder, but more brittle and have lower elasticity.

- Line 20: "teeth" or "tooth-like" elements as reported in line 52?

We stick to "teeth" for readability. Conodont teeth are not homologous with the teeth of other vertebrates, but so are eyes of arthropods and vertebrates or wings of pterosaurs and insects; yet they are still referred to as "eyes" and "wings". We added the clarification "conodont teeth, which evolved in parallel to all other vertebrate teeth" at the beginning of the introduction to make sure there is no misunderstanding.

- Lines 35-37: "Vertebrates are ... the one that has achieved the highest diversity and

disparity": What about insects? As regards disparity, it is a delicate topic: disparity is not only morphology, but disparity can be any phenotypic aspect including behaviour and ecology.

- This was indeed not clear. We removed "disparity" as the term has several meanings and in terms of species richness vertebrates are not the most diverse group. However, we keep "disparity" because, compared to insects, vertebrates clearly dominate in terms of the range of environments (from deep marine active swimmers through arctic birds and humans landing on the Moon) and of body sizes.

- Line 45: "first vertebrates": the manuscript deals only with conodonts (six specimens ranging in age from the Cambrian to the Devonian). In that time frame many other vertebrates appeared in the sea and were going to jump on the land. Why wasn't that material investigated?

We tested the hypothesis by Donoghue (2001) about the FIRST vertebrate dental tissues. The appearance of the teeth precedes the emergence from the sea to the land by hundreds of millions of years and there is no causal link between these events. In fact, many toothed vertebrate lineages never left the oceans. We focus on the origins of the first dental tissues and hope that the scientific community will endorse and adapt our methodology to study further questions on teeth evolution.

In general:

- What about the role of diagenesis on your material?

The revised version has been expanded with a thorough Raman analysis and discussion of possible influence of diagenesis.

- You base your conclusions on a very few specimens (six): how confident can you be on such a limited set?

- All quantitative analyses (Texture Index, parameters of the ν_1 - PO_4^{3-} band carried out using Raman spectroscopy) and all our conclusions are based on statistical tests carried out at the confidence level of 0.05. For example, the differences in TI between taxa are so big that they are significant even for such a small number of samples.

Reviewer #3 (Remarks to the Author):

3.1 Here, the authors did not consider any taphonomic status of each specimen. I felt that the authors treated the specimens as if they were neontological specimens, which are free from taphonomic consideration. For example, *Proconodontus muelleri* is Cambrian in age, and *Palmatolepis* sp. is Devonian in age, so there is nearly 100 Myr time gap between the two specimens. So one may expect that 'biogenic' minerals in Cambrian *Proconodontus muelleri* is comparatively less well-preserved than those of Devonian *Palmatolepis* sp. and it may influence the difference between the crystallographic alignments of two specimens. But there is no consideration like this in the current study.

Besides, even if ages of two specimen are similar (e.g. Devonian *Tripodellus* and *Palmatolepis*), spatial setting can affect the preservation. For example, when a formation of conodont A was deeply buried or under the influence of significant tectonic pressures and heat, whereas conodont B was buried in a 'peaceful formation', which was relatively free

from pressure and heat, the preservation status of conodont A and B may not be the same. In my opinion, neglecting the taphonomic status of specimens can significantly undermine the validity of this study. Please note that consideration of taphonomic status is an important topic for many paleontological studies (e.g., see a great example of Casella et al. 2018, *Palaeo3*, <https://doi.org/10.1016/j.palaeo.2018.03.011>).

In the interest of brevity, we did not discuss diagenesis in depth in the first submission. Following the recommendations of reviewers, we have expanded on this aspect, adding more background about our material and Raman spectrometric analyses comparing our specimens with known parameters of diagenetically altered bioapatites. This aspect is now a substantial part of the manuscript, but these analyses demonstrated that our selection of specimens was successful: Raman spectrometry indicated very homogenous parameters among our six specimens. This homogeneity indicates that ultrastructural differences we describe are not due to different diagenetic pathways.

3.2 Although I fully understand getting EBSD maps from conodonts is the very hard task, I still have concern for the small sample size (i.e., 6). There are many families in conodont (<https://en.wikipedia.org/wiki/Conodont>), so I expect that the number of genera is larger than that of families. So 'six' may be an insufficient number to represent the pattern in conodont evolution and it may weaken the robustness of the conclusion. In other words, a future study that uses much more diverse conodont taxa may report a different conclusion, and the current study is vulnerable to this type of challenge. In addition, if I understand correctly, just a single map was acquired from each specimen?

Bioapatite's sensitivity to the electron beam is the reason why there are so few EBSD studies of vertebrate tissues in general: not only of conodonts, but also of bones, teeth, fish scales etc. The acquisition of one map results in a complete destruction of the surface layer of the sample. For this reason, it is not possible to acquire more than one map from the same surface. In some cases we have re-ground, re-polished and re-analysed the same surface if the initial EBSD analyses did not yield sufficient signal. This illustrates how difficult to obtain this data was, hence we base our analysis on six samples, carefully selected to represent a wide range of feeding modes and evolutionary steps.

In my experience, the crystallographic orientation (i.e. pole figure) is pretty local (especially a-axis direction), so if a different part of the same specimen is analyzed with EBSD, the results (e.g. pole figure) may be different. In other words, just a single map per a specimen may not guarantee the solid reproducibility, which is required for Nat. Comm. papers.

The areas represented in EBSD maps are not selected at random, but target areas that are functionally and (in a wider sense) biologically homologous: the occlusal surfaces (which perform biting). We did acquire data from other areas in these specimens and all this data is made available as supporting material for the study (also for the reviewers), but comparing orientation between non-homologous functional surfaces would be meaningless since these surfaces have no evolutionary reasons to have similar material properties. Within these areas, we recalculated ODF and all indices (TI, M-index, pfJ) for random subsets to eliminate the bias of the analyzed area of the sample.

Again, please do not get me wrong. I truly think 'six' is already a great accomplishment, but I just feel that it is somewhat weaker than the 'level of support' criterion of Nat. Comm.

Please see our response to Reviewer 2: All quantitative analyses (Texture Index, parameters of the ν_1 - PO_4^{3-} band carried out using Raman spectroscopy) and all our conclusions are based on statistical tests carried out at the confidence level of 0.05. For example, the differences between TI are so big, that they are significant even for such a

small number of samples.

I just wish that the contribution of this study and that of Atakul-Özdemir et al. (2021, R. Soc. Open Sci.) should have been clearly explained in the main text to guarantee the significance of this study.

An explanation has been added in the Discussion.

5. Data and methodology

5.1 About the methodology: The panel B of Extended Data Figure 2 used a 'cleaning' to enhance the mapping. However, compared to the panel A, the cleaned maps in the panel B look 'too clear' and it may mean that the authors used too radical cleaning method for the panel B.

If this cleaning method was used in Figure 1, I would say that Figure 1 may contain pretty much 'false-positive' data, which are attributable to the radical cleaning. If Figure 1 used 'raw data', which was not influenced by any cleaning, please dismiss my current comment (and I believe the authors did not apply 'radical cleaning' to Figure 1).

Anyway, please provide this sort of information and all other steps applied to present Figure 1 to guarantee the full reproducibility. Currently, I felt that the method section is not detailed enough, so some steps could not be evaluated with enough confidence.

We have added extensive explanations in the text which should make it clear that no cleaning has been applied. Please note that all analyses were carried out using MTEX for MATLAB and we provide all code used for the analysis, which allows anyone to reproduce the study.

7. Suggested improvements

7.1 Taphonomic consideration: I strongly recommend that Casella et al. (2018) be a helpful literature that explains why considering taphonomic status is important for biomineral studies. There, one can see that taphonomic influences degrade the calcite in brachiopods and it can make 'fake news' from the old past. Admittedly, calcite is soft mineral as authors pointed out, so bioapatite fossils like conodont may not be severely affected by diagenesis (against my concerns). However, at least investigating the taphonomic status of the materials used in this study may be able to defend the concerns of mine and potential future readers.

I suggest that investigating the Raman spectra of conodont may yield useful information at least for thermal alteration. Please see McMillan and Golding (2019, Palaeo3) for details. If it turns out that Cambrian materials are heavily thermally influenced while younger Devonian materials (the ones used in Figure 1) are less affected by thermal alteration, I think this factor may contribute to 'false-positive' pattern (highly aligned Devonian signals vs weakly aligned Cambrian signals) in Figure 1. So I wish that the authors eliminate this sort of possibility completely with additional Raman data.

At the same time, I know Raman spectroscopy is not always available, and learning new skills used in McMillan and Golding (2019) can be a harsh request as a revision. So I think somewhat more practical suggestion (though I prefer Raman data) would be to investigate the classic conodont Color Alteration Index of the materials used in this study. If CAI shows that older materials are not always more heavily thermally altered, I believe the pattern reported in this study can be more safely understood as the biological signal as authors claimed (against my "devil's advocate" concern).

We have added a Raman spectroscopic analysis (and included a specialist among the authors) and provided CAI values for all samples. We also discuss the comparison with studies by Casella et al. (2018).

8. Clarity and context

The text is generally clear and explanation of context at Introduction is excellent (another A+). But some parts are hard to follow. Please see my further comments in “11. Additional comments”. Also, I believe the title tries to encompass too wider implications. In my opinion, even if the title is simply constrained to conodont instead of wider ‘early vertebrate’, this study is equally awesome, so please consider this issue during the revision.

We have changed the title to accommodate this comment.

11. Additional comments (about writing)

11.1 Title

In my opinion, it needs to be more focused on the core contents. When I read the title, I anticipated that this study used wide variety of vertebrates fossils (especially diverse Agnatha of Paleozoic) or at least analyzed much more diverse taxa of conodont. Please do not get me wrong. I strongly believe that getting the crystallographic data from conodonts is a very important contribution, but current title may not be representative for the contents of the paper because all results were obtained from the conodonts, and someone may think that just one clade (=conodont) is not enough to represent ‘early vertebrate’. Although I know that a concise title is preferred for a journal like Nat. Comm., I still believe that the best title should be a direct indicator for the contents.

The title has been changed.

11.2 Abstract

It is a well written summary. I point out two things: 1. There are four citations, but to my knowledge, Nat. Comm. does not allow citations in abstracts; 2. It is composed of more than 150 words, which is a length limit for abstracts of Nat. Comm.

I feel the current abstract is more like a that of Nature, so please revise the abstract to meet the guideline of Nat. Comm.

This was a result of an automatic manuscript transfer, to which we agreed. We changed the title to match Nat. Comms. requirements.

11.3 Introduction

The background information is very well presented and justifies the research. However, the arrangement needs improvements. The findings of other research and the contribution of current research is somewhat jumbled. I feel clear separation of ‘what others did (i.e., literature review)’ and ‘what we did in this research (i.e., Here we, blah blah)’ will make a better Introduction.

In the interest of keeping the introduction crisp and concise, we do not review previous research on conodont evolution and biomineralization in the Introduction. We cover this broader context of the study in the discussion. However, we now clarify in the Introduction that our study is a test of the hypothesis proposed by Donoghue (2001), before methods for testing it were available.

The authors also mentioned c-axis of apatite and it is an essential terminology for the study. My small concern is that very few readers will know what ‘c-axis’ means. I would recommend that the authors prepare a simple supplementary figure to show a unit cell apatite and its axes direction. I believe it will help readers to follow your thoughts. Please see also my minor comments in the annotated PDF

A new figure (now Fig. 1) has been added to clarify it.

11.4 Methods (Preparation and data acquisition)

If space permits, please briefly explain how this study succeeded in getting clear Kikuchi bands from the polished surface of conodont unlike earlier research, which were not that successful. Although Shirley et al.10 was cited, readers may want to know the difference without visiting Shirley et al.10.

Please see also my minor comments in the annotated PDF.

This is now included in the text.

11.4 Extended Data

(1) Figure 1. Although SEM images are very good at showing the morphological details, I recommend stereoscopic microscope images (e.g. Panels D, E) because they show color. I think color image is especially important (please see my “7.1 Taphonomic consideration” above). If possible, please provide.

We provided light microscopy photos of specimens for which they had been taken.

(2) Figure 2(a). It is very hard to find any difference among the different grain boundary thresholds. It is understandable because just one degree of difference is very subtle, but I am not sure then what angle is the most suitable for the match between ‘human eye’ and grain boundary threshold. I anticipated the answer from the authors somewhere in Lines 441–454, but there was no answer (also in Lines 135–138). It may confuse readers. Figure 2(b). Because there is no detailed information for the “standard grain smoothing and filter (here half-quadratic)” (Line 446), I cannot provide clear feedback. However, if ‘Filled grains’ maps are the results of ‘cleaning (=extrapolation of pixels)’ of the original data presented in Panel (a), I think it is problematic because the ‘cleaning’ step was too aggressive and radical. As authors pointed out, the ‘boundaries’ in the cleaned maps may not show the precise positions of real boundaries, grain shapes, and sizes. Most importantly, orientation information may be affected by this aggressive, anthropogenic factor. Did the authors used the “standard grain smoothing and filter (here half-quadratic)” cleaning method to make Figure 1 (I hope not)?

We did not clean any of the data in this study. This is now clearly laid out in the text. The code for the analyses is provided and allows reproducing the results.

11.6 Results

(1) Lines 118–122: Here, the description is very hard to follow. What are the ‘elongated apatite grains’? There are many shapes of grain in SIX EBSD maps, so it’s hard to know what I should see. In addition, due to the diverse shapes of apatite, I am pretty skeptical that the description is generally applicable to the maps in Figure 1.

Indeed, the elongation is not clearly visible in all maps, therefore this sentence has been removed.

(2) Figure 1: Although the maps are most probably ‘inverse pole figure maps’, no legend was provided so it is hard to fully understand the meaning of colors in the maps. In addition, there is no explanation for the numbers above the pole figures. I believe they are MUD (multiple of uniform density) and if it is correct, please explain at the caption.

This has been added and details in the caption and text have been added. Inverse pole figures are now explained in the Methods.

11.7 Discussion

(1) Lines 197–202: Similar to Lines 118–122, the authors keep mentioning ‘elongated grains’, but it is not that clear to readers. Please clarify what are ‘elongated grains’ and get more generality for this observation.

Given that we propose that grain reconstruction is not applicable in this case, we could not quantify the shape of the grains. It is, however, not a key finding in this study, therefore we removed the mentions of grain shapes to focus on the main points that can be clearly seen in the figures.

Reviewer #4 (Remarks to the Author):

1. How did you rule out the influence of diagenesis on the development of preferred crystallographic orientations of conodont fossils? You mentioned that “the hyaline tissue is widely used by geochemists thanks to its outstanding resistance to diagenesis (lines 67-68)”, but this does not guarantee the resistance of recrystallization. The related reference (15. Zhang et al., 2017) also does not suggest that recrystallization of conodont apatite does not occur easily by diagenesis, but suggests the integrated approach to recognize the nature and degree of diagenetic alteration in conodont specimen. Is there any other evidence that development of preferred crystallographic orientations of biomineral did not be affected by diagenesis? In this study, the spatial distribution of conodont samples is varied, such as U.S.A., Ukraine, Germany, Sweden, Canada, and Belgium (maybe?). The time variation is also large, from upper Cambrian to upper Devonian. These may indicate that the environments of diagenesis of samples should be different. That is, how did you rule out the effect of pressure and temperature on the crystallographic texture?

Please see our response to similar questions in previous reviews:

The revised version has been expanded with a thorough Raman analysis and discussion of possible influence of diagenesis.

In addition, you mentioned that “the discordance between the long axes of grains and the c-axes is unusual in abiotic minerals but common in biological tissues (line 121-122)”. What are the long axes of elongated apatite grains? From what I understand, the long axis of apatite is the direction of maximum length of same colored grain in grain color maps within Figure 1. In Fig.1f, the elongated long axes of apatite are likely to be perpendicular to the crystallographic c-axis of apatite. However, this trend is rarely observed in other samples. In Fig.1a and 1b, the crown appears to consist of almost a large single crystalline grain. Especially, in Fig.1d, the elongated long axes of apatite are parallel to the c-axes of apatite. In this case, is there a possibility of recrystallization in these samples?

As mentioned in previous reviews, the shape of grains was not obvious across all specimens and we did not quantify it for reasons discussed in the manuscript. Therefore, we removed the references to grain shape, as they would indeed require a thorough analysis in a separate study. The large single crystallite mentioned by the reviewer is linked to the fact that conodonts have two types of crown tissues, hyaline and white matter, with the latter characterized by larger crystals than the former. However, the cutoff between these two types is still debated and there is criterion agreed upon where one type ends and another starts. Since this is not essential for the conclusions of our study, we omit the references to grain shape and sizes in the revised version.

2. In this study, six hyaline tissues of conodont dental elements were analyzed for six taxa of

conodonts. That is, only one section of hyaline tissue was selected for each taxon. In this case, each specimen was understood to be representative of each taxon of conodonts.

However, is the cross-section representative sufficient to interpret the skeletal evolution of conodonts according to changes in feeding function? Of course, this result is the first successful analysis of EBSD for bioapatite, but I think that only one section for each taxon would not be enough to be representative of one taxon of conodonts. Although *Pro. muelleri* and *Pan. equicostatus* have a single coniform element in each, the other four samples appear to have multiple hyaline tissues. Are there similar trends in EBSD results for other hyaline tissues in each specimen?

To make the study maximally comparable, we show examples of functionally analogous organs: single piercing units in each type of element. Including entire elements would require discussing their occlusion and ontogeny, two aspects that are highly speculative and only familiar to a very narrow body of conodont specialists in the world. Given how speculative the interpretation would be, we decided against including the entire complexity and to focus on directly comparable organs. However, we provide the entire EBSD datasets collected from conodont elements, which allow reproducing the study and analysing the entirety of elements. An example of a comparison between denticles in one of the studied specimens, *W. excavata*, is now shown in Extended Data 10.

1. In Fig.1, what is the meaning of the reference frame (X- and Y-directions)? Do the black areas in schematic diagrams indicate the grain color maps in each specimen? From what I understand, you want to show the crystallographic c-axes of apatite grains aligned perpendicular to the occlusal surfaces (that is, aligned parallel to the occlusal (biting) axes). For this purpose, it is recommended to use an inverse pole figure (IPF) map in terms of the occlusal axis.

Of course, Fig.1 can show the c-axes aligned parallel to the occlusal surface, but an IPF or IPF map is generally used to show where the crystallographic axes of minerals are aligned in a specific direction (like in the reference 26. Kilian and Heilbronner, 2017). Because you processed the EBSD data using MTEX, a MATLAB toolbox, I believe that you can rotate the Euler angles to the specific direction. If X- and Y-directions are the sample coordinates in the SEM, you can make the X-direction aligned parallel to the occlusal surface, and Y-direction aligned perpendicular to the occlusal surface (that is, parallel to the occlusal axes). Then, when you plot the IPF map or IPF in terms of the Y-direction, it is easy to see which axis is perpendicular to the occlusal surface (in this case, maybe c-axis).

Otherwise, please indicate the meaning of the X- and Y- directions and match the directions between the schematic diagram and grain color map in each specimen.

We have followed the reviewer's recommendation and re-oriented the coordinates in the maps and added extensive explanation on how it was done.

2. You calculated the texture index (TI) to quantify the spread of crystallographic orientation of biogenic apatite. You tried many re-calculations of ODF and TI for random subsets to eliminate the bias of the analyzed size of sample, but you excluded the calculation of grain size because of no existence of known misorientation angle to discernible grain boundary in enamel and enamel-like tissues.

Of course, it is not easy to define the reference angle of grain boundary. However, in materials science (generally used for alloys), boundary with misorientation angle of $\sim 15^\circ$ is used to define a high-angle grain boundary (indicating the general grain boundary), and boundary with misorientation angle less than 15° is considered as a low-angle grain boundary (indicating the subgrain boundary). In the field of geology, this angle is considered

to be about 10° in the rock sample. In my opinion, the reference angle of the grain boundary for biogenic apatite would be similar to that of abiogenic (abiotic) minerals because of the similar mechanical properties between biogenic and abiogenic minerals. If you calculate the grain boundary of the sample, you can easily interpret the trend of TI values considering a grain size distribution.

TI values of *T. gracilis* and *Palmatolepis* sp. are higher than those of *Pro. muelleri* and *Pan. equicostatus* because former two samples are composed almost large single crystalline texture. The narrow TI value ranges of *Pro. muelleri* and *Pan. equicostatus* indicate the homogeneous grain size distribution. I don't know why grain size distribution is differ among the samples, but the crystallite of hyaline tissue appears to generally tend to become larger towards the later taxa of conodonts.

We could not accommodate this because it was not possible to reconstruct grains (see Extended Data 2).

3. In lines 93-95, are reference numbers 28 and 29 correct for ODF? Please check and correct with appropriate references (maybe Bunge, H.-J., 1982. Texture Analysis in Materials Science: Mathematical Models. Butterworths, London, UK).

In lines 100-101, a texture index (TI) also needs a citation. Please cite the appropriate references (probably Bunge (1982), or generally used as “J-index” by Mainprice and Silver (1993)).

We included these references as suggested.

If you want to just quantify the sharpness of pole figure of c-axes, pole figure J-index (pfJ-index) is also considerable. This value is analytically defined in a similar manner of J-index (Michibayashi and Mainprice, 2004), but as far as I know, there are no code in MTEX unfortunately. If you want to quantify the intracrystalline misorientation within a hyaline tissue, it is considerable to use a “misorientation index (M-index)” calculated from the misorientation angles among the uncorrelated grain pairs (Skemer et al., 2005). Unlike J-index (texture index), M-index has an advantage that it is hardly affected by the number of data, but also has a disadvantage of loss of spatial and textural information of the original dataset. I'm not sure whether you can get more reasonable result than J-index, but it is worth considering the M-index to quantify the crystallinity of hyaline tissue. However, be careful if you calculate the M-index by MTEX code. You have to do it directly using raw EBSD data, not using the result of ODF calculation.

We have now included the M-index and pfJ-index in the article and compared their ability to resolve different textures. We did not find substantial differences between these indices in terms of how their values characterize our specimens. Plots of the values of these three indices for all specimens are in the Extended Data 5 and 6. The Matlab code is provided for the calculation of all of them.

Reviewers' Comments:

Reviewer #1:

Remarks to the Author:

I am happy with the revised version and the modifications made by the authors.

Still, I would like to shortly comment on one point: in their rebuttal, the authors write: "Derived conodonts are those showing more specialized morphologies. Thus, more complex and more derived morphologies can co-occur with simpler and less derived ones: they are still separated by a long evolutionary distance."

Yet, simpler conodonts may descend from more complex ones, in particular it is not excluded that coniform elements, for instance from the Triassic (the example I was mentioning), are derived from more complex, platform conodonts. In that case evolutionary distance is not a simple function of morphological complexity as suggested by the authors.

Best regards

Nick

Reviewer #4:

Remarks to the Author:

Review to the manuscript titled "Increasing control over biomineralization in conodont evolution"

The work documented by Shirley et al. suggested the evolution of the first mineralized vertebrate tissues in conodont by progressive adaptation to dental function. The indexing of crystallographic axes of hydroxyapatite in conodont fossil by EBSD analysis was successful and the results well showed that c-axes of apatite consisting of hypermineralized crown tissues were aligned parallel/subparallel to the occlusal (biting) axes. It is interesting that the preferred crystallographic orientations of conodont's crown tissues could be developed by evolving conodont feeding mechanisms. Overall, the results of this manuscript will be interesting to geologists, especially mineralogist, and I think it may have the potential to provide important inspiration to paleontologists.

In addition, several issues for further consideration about diagenesis, texture index, and limited analyzed area are well covered in this manuscript. I recommend proceeding with the publication after addressing some minor typos, fixing mismatched data, and incorporating some noteworthy comments. Below is a summary of comments, not including all minor comments. Please see the annotations in attached PDF file.

1. Please check the <11-20> and <10-10> in Figs.1 and 2. If you want to show the <11-20>, you have to change the <10-10> figures. Detail comments are included in PDF file and code review.
2. I recommend matching the order of taxa in the legend for all figures (following Fig.3). Additionally, please check whether the color of the data and the legend are appropriately matched in Extended data 5.
3. The TI values seems to be different between Fig.3 and Extended data 5 and 10 (especially Pan. eucicostatus and Palmatolepis sp.). Please check these graph and data.

Please refer to the attached PDF file for additional details.

Reviewer #5:

Remarks to the Author:

This is a very interesting study that can contribute, not just to test the hypothesis about

biomineralization and functional morphology in conodonts, to extend biomineralization studies within the fossil record. This would rather depend on fossil preservation and this is an area in which authors can provide a better justification, without having to acquire much more additional data (the specific explanation below). On the other hand, authors do a good job addressing reviewers' comments in general, although a better explanation has to be provided to the third reviewer's concern about diagenesis.

Specific Comments to Authors:

- There is always a concern that diagenesis may be masking the "true value" of the provided EBSD data, even by itself is quite compelling in the manuscript for the color-coded crystallographic maps and pole figures. Yet, the Extended Data 9 (Band Contrast Maps) may indicate some level of alteration for elements in Fig. 9e and 9f. Knowing that the quality of these maps is depending on instrument type, analysis conditions, and step size, authors could provide some additional data to corroborate the 'good preservation' of such conodont elements. High-resolution band contrast maps tend to reveal some aspects of the microstructure and thus, it is useful to pair SEM images of the microstructure (admitting that sample preparation could potentially induce some level of dissolution and/or distortion of microstructural elements) with EBSD data. I have not been able to find any SEM images of the microstructure and I would suggest authors to provide this data.

- Authors provide additional Raman data to address the issue of diagenesis. In my view, this is the best approach, yet, I think authors do not use the Raman results in the best possible way. I think the FWHM of ν_1 -PO₄³⁻ is not the best way to prove their point as the potential variability of such value for that peak could be within error. A better way is to use the difference between the vibrational band peaks ν_3 and ν_4 . The adding value to this is that it can provide an additional measurement of crystallinity, reinforcing the EBSD data beyond addressing aspects related to diagenesis. I suggest authors to check the paper by Amini et al. 2020, ('Sharp-preserving erosion controlled by the graded microarchitecture of shark tooth enameloid', published in Nature Communications) to find out how to do this additional data analysis.

Response to referee comments to the article:

Reviewer comments in italics:

Reviewer #1 (Remarks to the Author):

I am happy with the revised version and the modifications made by the authors. Still, I would like to shortly comment on one point: in their rebuttal, the authors write: "Derived conodonts are those showing more specialized morphologies. Thus, more complex and more derived morphologies can co-occur with simpler and less derived ones: they are still separated by a long evolutionary distance."

Yet, simpler conodonts may descend from more complex ones, in particular it is not excluded that coniform elements, for instance from the Triassic (the example I was mentioning), are derived from more complex, platform conodonts. In that case evolutionary distance is not a simple function of morphological complexity as suggested by the authors.

This comment has helped us understand the reviewer's position in the previous round of revision better: indeed, the manuscript suggested that morphological complexity and evolutionary distance are correlated. What we should have said is that evolutionary distance correlates with adaptations at any level, not necessarily morphological but e.g., ultrastructural, or even behavioral. So, for example, better prey acquisition strategies or teeth less prone to failure can allow reducing the morphological complexity without the taxon changing its food base. We have modified two sentences in the main text accordingly, shifting the focus from morphological to more broadly understood adaptations.

Reviewer #4 (Remarks to the Author):

I recommend proceeding with the publication after addressing some minor typos, fixing mismatched data, and incorporating some noteworthy comments. Below is a summary of comments, not including all minor comments. Please see the annotations in attached PDF file.

1. Please check the <11-20> and <10-10> in Figs.1 and 2. If you want to show the <11-20>, you have to change the <10-10> figures. Detail comments are included in PDF file and code review.

The figure and code have been corrected.

2. I recommend matching the order of taxa in the legend for all figures (following Fig.3). Additionally, please check whether the color of the data and the legend are appropriately matched in Extended data 5.

We now applied a uniform ordering of specimens across all figures, preserving the order as shown in Fig. 2. This means, Fig. 4 has been modified to change the order, as well as Extended Data 5. The code for the figures has been updated on GitHub.

3. The TI values seems to be different between Fig.3 and Extended data 5 and 10 (especially *Pan. equicostatus* and *Palmatolepis* sp.). Please check these graph and data.

This is due to the randomized approach in the function we have written and published along with this manuscript. Our function samples random subsets of the area and calculates an average TI value from across subsets. We introduced this to account for different sizes of the datasets. Thus, each time the function is ran, as is the case for the datasets compared in Fig. 3 and Extended data 5 and 10, it gives slightly different results, but they remain qualitatively consistent (i.e. the mean values for each taxon remain in the same position, if you were to rank them according to the mean, with each run). The inconsistency spotted by the reviewer between Extended data 5 and 10 is due to swapped colors in the legend of Extended Data 5, also noted by the reviewer. We have updated the code (which is publicly available on GitHub) to amend the wrong label of the plot. Please note that the original code used to generate the figure is publicly available under version control, so it can be verified that the plot showed correct data, only colors were filled in the wrong order.

I just reviewed the MTEX code and noticed that you plotted the pole figures for {10-10} and {0001} for apatite.

However, as I understand, you intended to plot the a-axis and c-axis, that is, <11-20> and <0001>. The {10-10} and {11-20} are differen plane in hexagonal symmetry as I mentioned in attached PDF file.

So, please change the code like below:

```
plotPDF(ebsd.orientations,Miller({1,1,-2,0},{0,0,0,1},ebsd.orientations.CS,  
'UVTW'),'contourf')
```

Actually 'UVTW' may not affect the pole figure, but this will show the 'axis' of crystal, not the 'pole' of the plane.

However, plot again after changing from {1,0,-1,0} to {1,1,-2,0} if you want to reperenst the pole figure of a-axis (similar to Extended data 10).

Corrected in the figure and in the code on GitHub.

Reviewer #5 (Remarks to the Author):

- There is always a concern that diagenesis may be masking the “true value” of the provided EBSD data, even by itself is quite compelling in the manuscript for the color-coded crystallographic maps and pole figures. Yet, the Extended Data 9 (Band Contrast Maps) may indicate some level of alteration for elements in Fig. 9e and 9f. Knowing that the quality of these maps is depending on instrument type, analysis conditions, and

step size, authors could provide some additional data to corroborate the ‘good preservation’ of such conodont elements. High-resolution band contrast maps tend to reveal some aspects of the microstructure and thus, it is useful to pair SEM images of the microstructure (admitting that sample preparation could potentially induce some level of dissolution and/or distortion of microstructural elements) with EBSD data. I have not been able to find any SEM images of the microstructure and I would suggest authors to provide this data.

We agree with this suggestion 100%. We should note that we examined each sample in Secondary Electron mode before undertaking any analysis. If any topography was present, we re-ground and re-polished the sample until the surface was perfectly smooth (which is why the number of specimens is limited, this process can take months). We also examined them in the BSE mode to evaluate changes in composition. All these images were entirely dominated by growth layers, so we did not use them to assess diagenesis, as we could not spot its effects in this type of imaging. As a result, BSE images were not taken systematically using a consistent protocol, as they did not convey data which we were interested in. Those images we have are added to the supplementary materials, so readers can assess on their own, but we do not see any signal related to diagenesis. However, we suggest that this does not strengthen the argument about the lack of diagenetic influence, but rather shows that this type of imaging is not a good proxy for diagenesis in bioapatite.

- Authors provide additional Raman data to address the issue of diagenesis. In my view, this is the best approach, yet, I think authors do not use the Raman results in the best possible way. I think the FWHM of ν_1 -PO₄³⁻ is not the best way to prove their point as the potential variability of such value for that peak could be within error. A better way is to use the difference between the vibrational band peaks ν_3 and ν_4 . The adding value to this is that it can provide an additional measurement of crystallinity, reinforcing the EBSD data beyond addressing aspects related to diagenesis. I suggest authors to check the paper by Amini et al. 2020, (‘Sharp-preserving erosion controlled by the graded microarchitecture of shark tooth enameloid’, published in Nature Communications) to find out how to do this additional data analysis.

This was a very inspiring suggestion, as it related to most recent developments in Raman spectroscopy of apatite group minerals. We have to point out that all of the conodont teeth are hydroxyapatite, rather than fluorapatite that is observed in shark teeth and described in the paper suggested by the reviewer. The recent paper supervised by one of us, Helen King, on orientation effects in hydroxyapatite shows that the bands used in the suggested paper are highly orientation sensitive within these spectra (Gemeri et al. *Journal of Raman Spectroscopy* 54.2 (2023): 159-170). This is consistent with experimental findings for natural hydroxyapatite samples, references for which can be found therein. Although relative intensities are different depending on orientation, the full width at half maximum and peak position, which we use here, should not change due to orientational effects. Therefore, for our samples the approach suggested by the reviewer produces a mixed signal that is also be influenced by the orientation of the crystals.

We would like to point out that the application of these parameters to studying diagenesis is very novel and there are just a few datasets offering comparison or validation, which we compiled here.

We have also updated the supplementary data with the full Raman spectra data, in addition to the previously provided data on peak width and position.

Additional corrections have been made based on the comments provided by reviewer in the attached pdf. These corrections were:

1. Adding a reference to Fig. 1 and modifying it so that arrows in Fig. 1b are more clear and explained in the caption.
2. Letters referring to sub-figures of Fig. 2 have been corrected.
3. References from the Methods chapter have been unlinked from the references in the main text and now have their own numbering.
4. One typo has been removed.
5. A scale bar has been added to Extended Data 1e.
6. The silhouettes in Extended Data 10 have been corrected, as previously *Palmatolepis* sp. and *T. gracilis* had been accidentally swapped.

One reviewer suggestion has not been applied: “by? I believe that this subheading should be displayed in phrase form rather than in sentence form.” with respect to the sub-heading “The crystallographic texture of conodont denticles is an adaptation to dental function”. We stand by the original version, which we think is easier to follow for the readers.

Reviewers' Comments:

Reviewer #4:

Remarks to the Author:

The authors have thoroughly revised the manuscript and have effectively addressed most of the comments.

I believe that this manuscript can be acceptable after minor corrections. Please check the annotations for mismatched figure numbers and typos in the PDF file.

In line 186,
Figure 2f -> Figure 2a

In lines 189 and 192,
I think some TI values outside the parentheses are strange, such as 'much smaller range of TI values 7 to 33 (mean 24)' and 'T. gracilis 37 to 78 (mean 53) and *Palmatolepis* sp. 35 to 89 (mean 64)'. Please change these like 'B. cf. *aculeatus* (14 to 56, mean 33) and *W. excavata* (13 to 67, mean 36)' in line 190, or add some prep.

In line 205,
~. Based on -> ~, based on

In line 206,
Fig.1 -> Fig.2

In line 212,
Please add this: (Extended Data 5 and 6)

In line 215,
Figure 4 -> Figure 4a

In line 216,
Figure 5 -> Figure 4b

In line 690,
As I previously mentioned, I recommend unifying the term between 'pfJ' in main text and 'pfTi' in the extended data. Or, please indicate here that 'pfJ' and 'pfTi' refer to the same one.

In Extended Data1,
You mentioned that you added scale bar in Extended Data 1e, but I cannot find one. Please check.

Reviewer #5:

Remarks to the Author:

Authors mentioned that they added additional data to the supplementary ("extended data"), but I cannot find it. Specifically, this is the relevant information:

- As a result, BSE images were not taken systematically using a consistent protocol, as they did not convey data which we were interested in. Those images we have are added to the supplementary materials, so readers can assess on their own, but we do not see any signal related to diagenesis.

- We have also updated the supplementary data with the full Raman spectra data, in addition to the previously provided data on peak width and position.

Response to reviewers

Reviewer #4

The authors have thoroughly revised the manuscript and have effectively addressed most of the comments.

I believe that this manuscript can be acceptable after minor corrections. Please check the annotations for mismatched figure numbers and typos in the PDF file.

In line 186,
Figure 2f -> Figure 2a

Corrected.

In lines 189 and 192,
I think some TI values outside the parentheses are strange, such as 'much smaller range of TI values 7 to 33 (mean 24)' and 'T. gracilis 37 to 78 (mean 53) and *Palmatolepis* sp. 35 to 89 (mean 64)'. Please change these like 'B. cf. *aculeatus* (14 to 56, mean 33) and *W. excavata* (13 to 67, mean 36)' in line 190, or add some prep.

Please note that these values are obtained by resampling from the same dataset to account for the size of the measured area. As discussed in the manuscript, the Texture Index is sensitive to the measured area, so we account for it by creating a bootstrap empirical distribution of value pairs Area vs. TI, which is shown in Fig. 3b. So the min, max and mean values vary slightly each time the code is run, but without affecting the statistical test and the conclusion. In the final submission, we provide a frozen set of values resulting from one run that was used to create all figures, tables and tests, and we have switched the production of the figure to R Software so that we can provide the code for reproducing it. The details of the file and code for producing the figure, the values reported in the text and the results of the test applied to the Texture Index are provided in the Supplementary Data and Supplementary Code, respectively.

In line 205,
~. Based on -> ~, based on

Corrected.

In line 206,
Fig.1 -> Fig.2

Corrected.

In line 212,
Please add this: (Extended Data 5 and 6)

Because we have been asked by the editors to rename Extended Data to Supplementary Figures and Tables, respectively, we corrected this accordingly.

In line 215,
Figure 4 -> Figure 4a

Corrected.

In line 216,
Figure 5 -> Figure 4b

Corrected.

In line 690,
As I previously mentioned, I recommend unifying the term between 'pfJ' in main text and 'pfTi' in the extended data. Or, please indicate here that 'pfJ' and 'pfTi' refer to the same one.

The naming has been unified and 'pfJ' is used throughout.

In Extended Data1,
You mentioned that you added scale bar in Extended Data 1e, but I cannot find one.
Please check.

That was a mistake on our side, it is now corrected.

Reviewer #4 (Remarks on code availability):

I just reviewed the MTEX code, and the authors have properly revised it.

Reviewer #5 (Remarks to the Author):

Authors mentioned that they added additional data to the supplementary ("extended data"), but I cannot find it.

The BSE images concerned here are included in the submission as 'Supplementary_Data.zip'. They are also available under <https://osf.io/yxjts/> where we could annotate them with metadata. Both forms of making this data available are listed in "Description of Additional Supplementary Files" included in the submission.